# A Sinkhorn-type Algorithm for Constrained Optimal Transport

## Abstract

Entropic optimal transport (OT) and the Sinkhorn algorithm have made it practical for machine learning practitioners to perform the fundamental task of calculating transport distance between statistical distributions. In this work, we focus on a general class of OT problems under a combination of equality and inequality constraints. We derive the corresponding entropy regularization formulation and introduce a Sinkhorn-type algorithm for such constrained OT problems supported by theoretical guarantees. We first bound the approximation error when solving the problem through entropic regularization, which reduces exponentially with the increase of the regularization parameter. Furthermore, we prove a sublinear first-order convergence rate of the proposed Sinkhorn-type algorithm in the dual space by characterizing the optimization procedure with a Lyapunov function. To achieve fast and higher-order convergence under weak entropy regularization, we augment the Sinkhorn-type algorithm with dynamic regularization scheduling and second-order acceleration. Overall, this work systematically combines recent theoretical and numerical advances in entropic optimal transport with the constrained case, allowing practitioners to derive approximate transport plans in complex scenarios. In addition, we extend the formulation of this work to partial optimal transport and propose a fast algorithm with practical super-exponential convergence.

## 1 Introduction

Obtaining the optimal transport (OT) (Villani et al., 2009; Linial et al., 1998; Peyré et al., 2019) plan between statistical distributions is an important subroutine in machine learning (Sandler and Lindenbaum, 2011; Jitkrittum et al., 2016; Arjovsky et al., 2017; Salimans et al., 2018; Genevay et al., 2018; Chen et al., 2020; Fatras et al., 2021). In this work, we focus on an optimal transportation problem with a combination of inequality and equality constraints. A typical example is an OT problem with one inequality constraint:

$$\min_{P:P\mathbf{1}=r, P^\top \mathbf{1}=c, P\geq 0} C \cdot P, \text{ such that } D \cdot P \geq t, \tag{1}$$

where $\cdot$ stand for entry-wise inner product, $C \in \mathbb{R}^{n \times n}$ is the cost matrix, $t \in \mathbb{R}, D \in \mathbb{R}^{n \times n}$ encode the inequality constraint, and $c, r \in \mathbb{R}^n$ are respectively the source and target density. As an illustration, in Figure 1, we plot the optimal transport plan between 1D distributions when the main cost is the transport cost induced by the $l_1$ Manhattan distance, but an inequality constraint is placed on the transport cost induced by Euclidean distance. One can see that solving inequality-constrained OT problems in equation 1 allows one to obtain transport maps under more complex geometric structures.

For the unconstrained OT problem, the most important recent breakthrough is the introduction of entropic regularization and the resultant Sinkhorn algorithm (Yule, 1912; Sinkhorn, 1964; Cuturi, 2013). With simple matrix scaling steps, the Sinkhorn algorithm gives an approximate OT solution in near-linear time (Altschuler et al., 2017), which fuels the wide adoption in the machine learning community. This prompts the following natural question:

> Is there an extension of the Sinkhorn algorithm to constrained optimal transport problems?

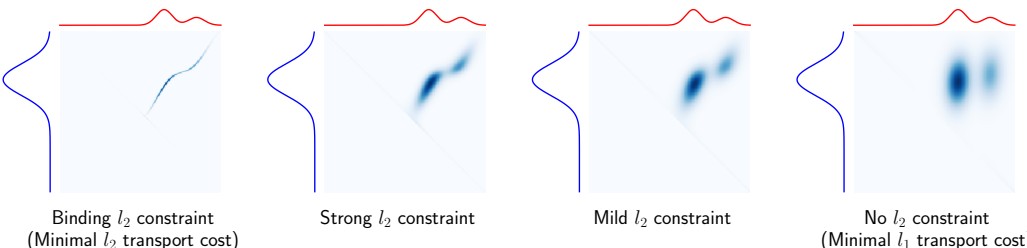

| Binding $l_2$ constraint
(Minimal $l_2$ transport cost) | Strong $l_2$ constraint | Mild $l_2$ constraint | No $l_2$ constraint
(Minimal $l_1$ transport cost) |

Figure 1: Illustration of 1D optimal transport under different inequality constraints, obtained with Algorithm 1. By tuning the inequality threshold, the transport plan evolves from minimizing the Euclidean distance transport cost to minimizing the Manhattan distance ($l_1$) transport cost.

This paper answers the above question in the affirmative. The key insight underlying the development in this work is that the entropic optimal transport problem is an instance of entropic linear programming (Fang, 1992), from which it is natural to extend entropy regularization to the constrained case. For an illustration, the example in equation 1 has the following relaxation:

$$\min_{P,s:P\mathbf{1}=r,P^\top\mathbf{1}=c,P\geq 0,s\geq 0} C \cdot P + \frac{1}{\eta}H(P,s),$$

$$\text{subject to } D \cdot P - s = t,$$

(2)

where $s$ is the slack variable, $\eta > 0$ is the entropy regularization parameter, and $H(P,s) := \sum_{ij} p_{ij} \log p_{ij} + s\log(s)$ is the entropy regularization term. The primal-dual form of equation 2 leads to a direct generalization of the Sinkhorn algorithm to the constrained case, which we develop in Section 2 and Section 3.

### 1.1 CONTRIBUTIONS

In addition to the introduction of a novel Sinkhorn-type algorithm for constrained OT, this paper systematically generalizes existing theoretical results as well as practical acceleration techniques from the existing entropic OT literature. In particular,

- Following the analysis in Weed (2018), we show that entropic optimal transport in the constrained case is exponentially close to the optimal solution.
- By extending the result in Altschuler et al. (2017), we show that a Sinkhorn-type algorithm reaches approximate first-order stationarity in polynomial time.
- For settings with weak entropic regularization, we further improve practical performance by introducing dynamic regularization scheduling in Chen et al. (2023) to the Sinkhorn-type algorithm, which features an adaptive entropy regularization term.
- We extend the approximate sparsity analysis in Tang et al. (2024) to the constrained OT case and introduce an accelerated second-order algorithm through sparse Newton iterations. We exhibit strong numerical evidence this technique outperforms conventional quasi-Newton methods.
- We introduce a variational formulation of the task, which naturally enables the use of first-order methods such as in Dvurechensky et al. (2018); Lin et al. (2019).
- We propose a novel numerical algorithm for partial optimal transport (Chapel et al., 2020) which exhibits practical super-exponential convergence.

### 1.2 RELATED LITERATURE

**Constrained optimal transport**  The numerical treatment for constrained OT focuses on special cases such as capacity constraints (Korman and McCann, 2013; 2015), multi-marginal transport (Gangbo and Swiech, 1998; Buttazzo et al., 2012; Benamou et al., 2015; Pass, 2015; Khoo et al., 2020), martingale optimal transport (Tan and Touzi, 2013; Beiglböck et al., 2013; Galichon et al., 2014; Dolinsky and Soner, 2014; Guo and Obłój, 2019), and partial optimal transport (Chapel et al.,

2020; Le et al., 2022; Nguyen et al., 2022; 2024). In terms of numerical methods for constrained OT, Benamou et al. (2015) defines an iterative Bregman projection approach for equality constraints and a Bregman–Dykstra iteration for general constraints. We show in Appendix I that the iterative Bregman projection is equivalent to our proposed update step in the equality case. Our contribution in relation to Benamou et al. (2015) is threefold. First, for equality constraints, we propose a generalized Sinkhorn-type algorithm that implements the iterative Bregman projection, whereas previous work only defines the projection step in special cases. Second, for inequality constraint, our approach is novel and enjoys theoretical guarantee as seen in our Theorem 1 and Theorem 2. Finally, the Bregman–Dykstra iteration in (Benamou et al., 2015) requires a simple closed-form solution for each projection step, whereas our proposed method applies to general cases.

**OT in machine learning**   A substantial amount of research literature exists on the use of optimal transport in different areas of machine learning. Notable applications include statistical learning (Kolouri et al., 2017; Vayer et al., 2018; Genevay et al., 2019; Luise et al., 2018; Oneto et al., 2020; Huynh et al., 2020), domain adaptation (Fernando et al., 2013; Redko et al., 2017; Courty et al., 2017; Alvarez-Melis et al., 2018; Nguyen et al., 2022; 2021; Xu et al., 2022; Turrisi et al., 2022), and using optimal transport distance in designing training targets (Genevay et al., 2017; Bousquet et al., 2017; Sanjabi et al., 2018; Deshpande et al., 2019; Lei et al., 2019; Patrini et al., 2020; Onken et al., 2021). The usage of entropic optimal transport in the constrained case allows practitioners to query transport plans which satisfy more complex structure than in the unconstrained case, which are beneficial across all the applications above.

**Acceleration for Sinkhorn**   There is a considerable body of work in speeding up the runtime of the Sinkhorn algorithm, and a significant portion of our numerical treatment is devoted to the extension of these algorithms to the constrained case. In addition to (Chen et al., 2023; Tang et al., 2024) to be discussed in Section 3, we mention a few noteworthy methods for acceleration. Randomized or greedy row/column scaling (Genevay et al., 2016; Altschuler et al., 2017) can be directly extended to the constrained case by the inclusion of the dual variables for the linear constraints. Likewise, methods based on the variational form of entropic OT, such as Nesterov acceleration (Dvurechensky et al., 2018) and mirror descent (Lin et al., 2019; Kemertas et al., 2023), can be used in the constrained case by considering the Lyapunov function introduced in Section 2. While the majority of existing techniques can be directly extended to the constrained case, one notable exception is the class of methods based on approximation to the kernel $K = \exp(-C\eta)$ (Deriche, 1993; Solomon et al., 2015; Bonneel et al., 2016; Altschuler et al., 2019; Scetbon and Cuturi, 2020; Lakshmanan et al., 2022; Huguet et al., 2023; Li et al., 2023). The main obstruction from performing kernel approximation in the constrained case is that the kernel is changing throughout the optimization process due to update in the constraint associated dual variables. As the kernel compression step is costly and often done in the offline stage, it poses a significant challenge if one has to update the kernel approximation with each dual variable update dynamically.

## 2   THEORETICAL FOUNDATIONS OF CONSTRAINED ENTROPIC OPTIMAL TRANSPORT

This section summarizes the entropic optimal transport in the constrained case.

**Notations**   The symbol $n$ is reserved for the system size of constrained OT problems, with the cost matrix satisfying $C \in \mathbb{R}^{n \times n}$. We use $M \cdot M' := \sum_{ij} m_{ij} m'_{ij}$ to denote the entry-wise inner product. For a matrix $M$, the notation $\log M$ stands for entry-wise logarithm, and similarly $\exp(M)$ denotes entry-wise exponential. We use the symbol $\|M\|_1$ to denote the entry-wise $l_1$ norm, i.e. $\|M\|_1 := \|\text{vec}(M)\| = \sum_{ij} |m_{ij}|$. The $\|M\|_\infty$ and $\|M\|_2$ norms are defined likewise as the entry-wise $l_\infty$ and $l_2$ norms, respectively. The notation $\mathbf{1}_{n \times n}$ is the all-one $n \times n$ matrix, and the notation $\mathbf{1}$ denotes the all-one vector of appropriate size.

**Background**   For simplicity, we assume the target and source density $r, c \in \mathbb{R}^n$ satisfies $\sum_i r_i = \sum_j c_j = 1$. For any inequality constraint of the form $D \cdot P \geq t$, note one can convert the condition to $(D - t\mathbf{1}_{n \times n}) \cdot P \geq 0$, and similarly in the equality case. Let $K$ and $L$ respectively denote the number of inequality and equality constraints. By the construction above, for each optimal transport

problem under linear constraint, there exists $D_1, \ldots D_K, D_{K+1}, \ldots, D_{K+L} \in \mathbb{R}^{n \times n}$ so that the linear constraints are encoded by $\mathcal{I}, \mathcal{E}$ where

$$\mathcal{I} := \bigcap_{k=1,\ldots,K} \{P \mid D_k \cdot P \geq 0\}, \quad \mathcal{E} := \bigcap_{l=1,\ldots,L} \{P \mid D_{l+K} \cdot P = 0\}. \tag{3}$$

We summarize the *general form of constrained optimal transport* by the following the following linear program (LP):

$$\min_{P: P\mathbf{1}=r, P^\top \mathbf{1}=c, P \geq 0} C \cdot P, \text{ such that } P \in \mathcal{S}, \tag{4}$$

where $\mathcal{S} := \mathcal{I} \cap \mathcal{E}$ for $\mathcal{I}, \mathcal{E}$ defined in equation 3.

**Entropic optimal transport under constraint**  Under entropic linear programming (Fang, 1992), one can write down the formulation for entropic optimal transport under general linear constraints. For $k = 1, \ldots, K$, we define $s_k$ to be the slack variable corresponding to $D_k \cdot P$. The *constrained entropic optimal transport* follows from the following equation:

$$\min_{\substack{P,s: P\mathbf{1}=r,\, P^\top \mathbf{1}=c, \\ P \geq 0,\, s_k \geq 0\, \forall\, k}} \quad C \cdot P + \frac{1}{\eta} H(P, s_1, \ldots, s_K),$$

$$\text{subject to} \qquad D_k \cdot P = s_k \text{ for } k = 1, \ldots, K. \tag{5}$$
$$D_{l+K} \cdot P = 0 \text{ for } l = 1, \ldots, L,$$

where the entropy term is defined by $H(P, s_1, \ldots, s_K) = \sum_{ij} p_{ij} \log(p_{ij}) + \sum_{k=1}^{K} s_k \log s_k$.

We motivate the optimization task in equation 5 by Theorem 1, which shows that the entropy-regularized optimal solution is exponentially close to the optimal solution:

**Theorem 1.** *For simplicity, assume that $\sum_{i \in [n]} r_i = \sum_{j \in [n]} c_j = 1$, the LP in equation 4 has a unique solution $P^\star$, and assume that $\|D_k\|_\infty \leq 1$ for $k = 1, \ldots, K$. Denote $P_\eta^\star$ as the unique solution to equation 5. There exists a constant $\Delta$, depending only on the LP in equation 4, so that the following holds for $\eta \geq \frac{(K+1)(1+\ln(4n^2(K+1)))}{\Delta}$:*

$$\|P_\eta^\star - P^\star\|_1 \leq 8n^{\frac{2}{K+1}}(K+1) \exp\left(-\eta \frac{\Delta}{K+1}\right).$$

The definition of $\Delta$ and the proof are deferred to Appendix F. In this work, we only consider examples for which $K = O(1)$, in which case the bound in Theorem 1 does not significantly differ from the unconstrained case. It should be noted that the dependency on $n$ goes down with an increasing number of inequality constraints $K$, but one can check the bound is monotonically worse as $K$ increases. Interpreting the bound could mean that adding more inequality constraints limits the degree of freedom but may also potentially amplify the error caused by entropy regularization.

**Variational formulation under entropic regularization**  By introducing the Lagrangian variable and using the minimax theorem (for a detailed derivation, see Appendix H), we formulate the associated primal-dual problem to equation 5 as encoded by the primal-dual function $L$:

$$\max_{x,y,a} \min_{P,s} L(P, s, x, y, a) := \frac{1}{\eta} P \cdot \log P + C \cdot P - x \cdot (P\mathbf{1} - r) - y \cdot (P^\top \mathbf{1} - c)$$

$$+ \frac{1}{\eta} \sum_{k=1}^{K} s_k \log s_k + \sum_{k=1}^{K} a_k s_k - \sum_{m=1}^{K+L} a_m (D_m \cdot P), \tag{6}$$

where $s = (s_1, \ldots, s_K)$ is the shorthand for slack variable, $a = (a_1, \ldots, a_K, a_{K+1}, \ldots, a_{K+L})$ is the shorthand for the constraint dual variables (excluding the original row/column constraints).

By eliminating $P, s$ (see Appendix H), the function $f(x, y, a) := \min_{P,s} L(P, s, x, y, a)$ admits the following form

$$f(x, y, a) = -\frac{1}{\eta} \sum_{ij} \exp\left(\eta\left(-C_{ij} + \sum_{m=1}^{L+K} a_m (D_m)_{ij} + x_i + y_j\right) - 1\right)$$

$$+ \sum_i x_i r_i + \sum_j y_j c_j - \frac{1}{\eta} \sum_{k=1}^{K} \exp(-\eta a_k - 1). \tag{7}$$

As a consequence of the minimax theorem, maximizing over $f$ is equivalent to solving the problem defined in equation 5. We emphasize that $f$ is *concave*, allowing one to use routine convex optimization techniques.

Define $P = \exp\left(\eta(-C + \sum_m a_m D_m + x\mathbf{1}^\top + \mathbf{1}y^\top) - 1\right)$ as the intermediate matrix corresponding to dual variables $x, y, a$. We write down the first derivative of the Lyapunov function $f$ (also known as the dual potential function):

$$
\begin{aligned}
\nabla_x f &= r - P\mathbf{1}, \quad \nabla_y f = c - P^\top\mathbf{1}, \\
\partial_{a_k} f &= \exp(-\eta a_k - 1) - P \cdot D_k, \quad \forall k \in [K], \\
\partial_{a_{l+K}} f &= -P \cdot D_{l+K}, \quad \forall l \in [L].
\end{aligned}
\tag{8}
$$

Indeed, one can use any first-order method on $f$. Moreover, methods based on accelerated gradient descent have shown good practical performance in optimal transport. One notable example is the adaptive primal-dual accelerated gradient descent (APDAGD) algorithm, which has been shown to outperform the Sinkhorn algorithm during the initial stages of optimization (Dvurechensky et al., 2018). Appendix D details the extension of APDAGD to constrained optimal transport using the primal-dual form shown in equation 6. Section 5 shows that APDAGD likewise enjoys good numerical performance for constrained optimal transport. Overall, the APDAGD leaves room for improvement, as the numerical result in Section 5 shows that our main approach enjoys better convergence properties.

**Extension to partial optimal transport**   The main formulation of this work precludes the case of partial optimal transport (POT) Chapel et al. (2020). In particular, POT places inequality constraint for the row sum and column sum for the transportation matrix, which is a different problem setting than those considered in equation 5. We develop an extension of our approach to the case of POT, and in particular, we showcase a practical numerical algorithm with practical super-exponential convergence.

## 3   MAIN ALGORITHM

This section proposes an efficient Sinkhorn-type algorithm in the constrained case. We assume $K + L = O(1)$ to ensure the efficiency of the proposed approach.

**Main idea of the algorithm**   The variational formulation under entropic regularization shows that one can effectively solve for the entropic formulation in equation 5 by solving for the optimization task:

$$
\max_{x\in\mathbb{R}^n, y\in\mathbb{R}^n, a\in\mathbb{R}^{K+L}} f(x, y, a)
$$

for $f$ given in equation 7. We define a Sinkhorn-type algorithm by introducing three iteration steps:

1. ($x$ update) $x \leftarrow \arg\max_{\tilde{x}} f(\tilde{x}, y, a)$,
2. ($y$ update) $y \leftarrow \arg\max_{\tilde{y}} f(x, \tilde{y}, a)$,
3. ($a$ update) $a, t \leftarrow \arg\max_{\tilde{a}, \tilde{t}} f(x + \tilde{t}\mathbf{1}, y, \tilde{a})$

The proposed alternating update approach is summarized in Algorithm 1.

**Implementation detail**   Let $(x, y, a)$ be the current dual variables and define

$$
P = \exp\left(\eta(-C + \sum_m a_m D_m + x\mathbf{1}^\top + \mathbf{1}y^\top) - 1\right)
$$

as the current intermediate transport plan defined by these dual variables. By equation 8, one has $\nabla_x f = 0 \iff P\mathbf{1} = r$ and $\nabla_y f = 0 \iff P^\top\mathbf{1} = c$. Thus, the $x, y$ update steps amount to row/column scaling of the matrix $P$, which is identical to the Sinkhorn algorithm.

The $a$ update step constitutes the main novelty of our algorithm in the constrained case. In particular, the inclusion of the $t$ variable improves numerical stability by enforcing a normalization condition:

Suppose $P$ is the intermediate transport plan formed by the dual variable $(x + t\mathbf{1}, y, a)$, then

$$\partial_t f(x + t\mathbf{1}, y, a) = 0 \iff \sum_{ij} P_{ij} = \sum_i r_i, \tag{9}$$

which ensures that terms of the form $M \cdot P$ can be bounded by $\|M\|_\infty \left(\sum_i r_i\right)$. Thus, optimality in the $t$ variable ensures boundedness in the derivatives of $f$ such as those in equation 8.

We propose to use Newton's method for the $a$ update step. Namely, by directly computing the gradient and the Hessian, $\nabla_{at} f, \nabla^2_{at} f$, one uses the search direction $(\Delta a, \Delta t) = -\left(\nabla^2_{at} f\right)^{-1} \nabla_{at} f$. The learning rate is obtained through the standard backtracking line search scheme (Boyd and Vandenberghe, 2004).

---

**Algorithm 1** Sinkhorn-type algorithm under linear constraint

---

**Require:** $f, x_{\text{init}}, y_{\text{init}}, a_{\text{init}}, N, i = 0, \epsilon > 0$
1: $(x, y, a) \leftarrow (x_{\text{init}}, y_{\text{init}}, a_{\text{init}})$
2: **while** $i < N$ **do**
3:      $i \leftarrow i + 1$
4:      # Row&Column scaling step
5:      $P \leftarrow \exp\left(\eta(-C + \sum_m a_m D_m + x\mathbf{1}^\top + \mathbf{1}y^\top) - 1\right)$
6:      $x \leftarrow x + \left(\log(r) - \log(P\mathbf{1})\right)/\eta$
7:      $P \leftarrow \exp\left(\eta(-C + \sum_m a_m D_m + x\mathbf{1}^\top + \mathbf{1}y^\top) - 1\right)$
8:      $y \leftarrow y + \left(\log(c) - \log(P^\top\mathbf{1})\right)/\eta$
9:      # Constraint dual update step
10:     $a, t \leftarrow \arg\max_{\tilde{a}, \tilde{t}} f(x + \tilde{t}\mathbf{1}, y, \tilde{a})$
11:     $x \leftarrow x + t\mathbf{1}$
12: **end while**
13: Output dual variables $(x, y, a)$.

---

**Complexity analysis of Algorithm 1** The row/column scaling step is identical to the Sinkhorn algorithm and thus costs $O(n^2)$ per iteration. For the constraint dual update step, the computation cost is dominated by the calculation of $\nabla^2_{at} f$. By direct computation in Appendix C, one can show that the cost for obtaining the Hessian term $\nabla^2_{at} f$ is dominated by the computation of $\sum_{ij} P_{ij} (D_m)_{ij} (D_{m'})_{ij}$ for $m, m' = 1, \ldots, K + L$ and so the cost of the $a$ update step is $O((K + L)^2 n^2))$ per iteration of the Newton method. As the setting of this work assumes $K + L = O(1)$, one can see that each Newton step has a cost of $O(n^2)$. We set Algorithm 1 to run $N_a = O(1)$ Newton steps, as Newton's method enjoys super-exponential convergence practically (Boyd and Vandenberghe, 2004). Thus, the total cost of the $a$ update step is $O(N_a(K + L)^2 n^2)) = O(n^2)$.

### 3.1 ACCELERATION TECHNIQUES

**Entropy regularization scheduling** An important feature of the Sinkhorn algorithm is that the iteration complexity heavily depends on the entropy regularization term $\eta$, the tuning of which plays a significant part in practical performance. To that end, one can aid acceleration by using the doubling entropy regularization scheduling technique introduced in (Chen et al., 2023). For a desired entropy regularization value $\eta_{\text{final}}$, we take an initial regularization strength $\eta_{\text{init}}$ and take $N_\eta = \lceil \log_2(\eta_{\text{final}}/\eta_{\text{init}}) \rceil$. Then, one defines successively doubling regularization levels $\eta_0, \ldots, \eta_{N_\eta}$ so that $\eta_{\text{init}} = \eta_0 < \ldots < \eta_{N_\eta} = \eta_{\text{final}}$ and $\eta_i = 2\eta_{i-1}$ for $i = 1, \ldots, N_\eta - 1$. For each step $i$, one runs the subroutine in Algorithm 1 at $\eta_{i-1}$, and the obtained dual variable is used as the initialization when calling the subroutine for $\eta_i$.

**Second-order acceleration through sparsity** We further accelerate the Sinkhorn-type algorithm with Sinkhorn-Newton-Sparse (SNS) (Tang et al., 2024), a second-order method originally developed for the unconstrained case. Instead of alternating maximization as in Algorithm 1, one can instead introduce the combined variable $z = (x, y, a)$, where a naive strategy would be to optimize directly $f$ through a full Newton update of the type $z = z - \alpha(\nabla^2_z f)^{-1}\nabla_z f$, where $\alpha$ can be obtained through backtracking line search. However, a full Newton step would require an impractical $O((n + K + L)^3)$ scaling.

The SNS algorithm introduces a practical second-order method through a sparse approximation of the Hessian matrix, and we show how one can extend the algorithm to the constrained OT case. The key observation of SNS is that the Hessian submatrix corresponding to variable $x, y$ has a special structure:

$$\nabla_{xy}^2 f(x, y, a) = \eta \begin{bmatrix} \mathrm{diag}(P\mathbf{1}) & P \\ P^\top & \mathrm{diag}(P^\top \mathbf{1}) \end{bmatrix}, \tag{10}$$

which admits a sparse approximation as long as $P$ admits a sparse approximation. Moreover, the full Hessian matrix to $f$ admits a sparse approximation: As $K + L = O(1)$, the blocks of $\nabla^2 f$ corresponding to $\nabla_{xy}\nabla_a f$ and $\nabla_a^2 f$ lead to at most $O(n)$ nonzero entries to keep track of.

The rationale for approximate sparsity is simple: Under a mild uniqueness assumption, the optimal solution $P^\star$ to the LP in equation 4 has at most $2n - 1 + K + L = O(n)$ nonzero entries due to the fundamental theorem of linear programming (Luenberger et al., 1984). Moreover, the exponential closeness result in Theorem 1 implies that the entries of $P_\eta^\star$ decay at a rate of $\exp\left(-\frac{\eta\Delta}{K+1}\right)$ except at $2n - 1 + K + L = O(n)$ entries, which proves that approximate sparsity holds in the constrained case as well.

We propose an extended SNS algorithm by sparsifying the $\nabla_{xy}^2 f$ block in which one keeps only $O(n)$ nonzero entries. The sparse Newton step is used in combination with Algorithm 1 as a warm start to achieve rapid acceleration. The proposed scheme leads to the same $O(n^2)$ per-iteration complexity as SNS in the unconstrained case. Implementation details can be found in Algorithm 2 in Appendix C.

In addition, Appendix J includes the numerical performance of the Broyden–Fletcher–Goldfarb–Shanno (BFGS) algorithm and the limited-memory Broyden–Fletcher–Goldfarb–Shanno (L-BFGS) algorithm, which are two of the most widely used quasi-Newton methods. The numerical experiments shows that the two aforementioned quasi-Newton methods have inferior performance when compared to the sparse Newton iteration we propose.

## 4 CONVERGENCE ANALYSIS

In this section, we present a convergence bound for a modified version of the proposed Sinkhorn-type algorithm in terms of the stationarity condition. Similar to the proof in (Altschuler et al., 2017) for the unconstrained case, our proof strategy relies on characterizing convergence through the Lyapunov function $f$ introduced in equation 7. For the rest of this section, we assume $\sum_i r_i = \sum_j c_j = 1$.

We present a bound on the first-order stationarity condition. As one is performing alternating optimization for more than two variables, the convergence proof requires a modification to Algorithm 1, in which one examines the stationary condition on each variable and chooses the update step greedily. The following Theorem characterizes the approximate stationarity of the greedy version of Algorithm 1 (proof is in Appendix G):

**Theorem 2.** *Let $(x, y, a)$ be the current variable, let $P$ be the associated transport matrix, and let $d_m$ be as in equation 24. Define $Q_x, Q_y, Q_a$ by*

$$Q_x = \mathrm{KL}\left(r || P\mathbf{1}\right), Q_y = \mathrm{KL}\left(c || P^\top \mathbf{1}\right),$$

$$Q_a = \sum_{k=1}^{K} |d_k| \min\left(\frac{1}{8\eta}, \frac{|d_k|}{8\eta c_d + 4\eta(K+L)c_d^2}\right) + \sum_{l=1}^{L} \frac{d_{l+K}^2}{2\eta(K+L)c_d^2},$$

*which are respectively the right-hand-side of equation 22, equation 23, and equation 25.*

*Consider a greedy version of Algorithm 1, in which only one update step is performed at each iteration, and an $x$ (resp. $y$, $a$) update step is chosen if $Q_x$ (resp. $Q_y$, $Q_a$) is the largest among $(Q_x, Q_y, Q_a)$. Define $c_d := \max_{m \in [K+L]} \|D_m\|_\infty$, and define $c_g$ as the gap term in the Lyapunov function $f$ at initialization, i.e.*

$$c_g = \max_{\tilde{x}, \tilde{y}, \tilde{a}} f(\tilde{x}, \tilde{y}, \tilde{a}) - f(x_{\mathrm{init}}, y_{\mathrm{init}}, a_{\mathrm{init}}).$$

*Let $(x^u, y^u, a^u)$ be the dual variable after $u$ iterations and let $P_u$ be the associated intermediate transport plan. Then, the greedy algorithm outputs dual variables $(x, y, a)$ which satisfies the below approximate stationarity condition in $O(c_g \epsilon^{-2})$ iterations:*

$$\|\nabla f(x, y, a)\|_1 \leq \epsilon. \tag{11}$$

*Moreover, the matrix $P$ associated with the outputted dual variable satisfies the following condition:*

$$\epsilon \geq \|P\mathbf{1} - r\|_1 + \|P^\top \mathbf{1} - c\|_1 + \sum_{k=1}^{K} |\min(P \cdot D_k, 0)| + \sum_{l=1}^{L} |P \cdot D_{l+K}|, \tag{12}$$

*which shows that $P$ approximately satisfies the linear constraints in equation 4.*

## 5 NUMERICAL EXPERIMENTS

We conduct numerical experiments to showcase the performance of the proposed Sinkhorn-type algorithm and its acceleration techniques. Let $\mathcal{U}_{r,c}$ be the set of transport matrices from $r$ to $c$. We use the rounding algorithm in Altschuler et al. (2017) to obtain projection into $\mathcal{U}_{r,c}$. The performance is evaluated through the cost and constraint violation of the transport matrix one obtains through rounding. Specifically, one first uses the dual variable $(x, y, a)$ to form the intermediate matrix $P$. Then, one uses the rounding algorithm to get a transport matrix, denoted $\mathrm{Round}(P, \mathcal{U}_{r,c})$. The cost or score of the transport is evaluated as

$$\mathrm{Cost}(P) = C \cdot \mathrm{Round}(P, \mathcal{U}_{r,c}), \quad \mathrm{Score}(P) = (-C) \cdot \mathrm{Round}(P, \mathcal{U}_{r,c}).$$

For constraint violation, we use the following metric:

$$\mathrm{Violation}(P) = \sum_{k=1}^{K} |\min(\mathrm{Round}(P, \mathcal{U}_{r,c}) \cdot D_k, 0)| + \sum_{l=1}^{L} |\mathrm{Round}(P, \mathcal{U}_{r,c}) \cdot D_{l+K}|.$$

In Proposition 1 in Appendix G, we give an upper bound on $\mathrm{Violation}(P)$ when one runs the greedy version of Algorithm 1 in Theorem 2. Additional experiments can be found in Appendix A.

**Random assignment problem under constraints** In the first numerical test, we consider the random assignment problem (Mézard and Parisi, 1987; Steele, 1997; Aldous, 2001) with additional inequality and equality constraints. In this setting, we set the problem size of $n = 500$ and an entropy regularization of $\eta = 1200$. The source and target vectors are $c = r = \frac{1}{n}\mathbf{1}$. We consider three $n \times n$ matrices $C, D_I, D_E$, which respectively encode the cost, the inequality constraint, and the equality constraint. We generate the entries of $C, D_I, D_E$ by i.i.d. random variables following the distribution $\mathrm{Unif}([0, 1])$. We then set two threshold variables $t_I, t_E$ and consider the following optimal transport task:

$$\min_{P: P\mathbf{1}=r, P^\top \mathbf{1}=c} C \cdot P,$$
$$\text{subject to} \quad D_I \cdot P \leq t_I, D_E \cdot P = t_E. \tag{13}$$

The conversion of equation 13 to the general form is done by taking $D_1 = (D_I - t_I \mathbf{1}_{n \times n})/n$, $D_2 = (D_E - t_E \mathbf{1}_{n \times n})/n$. Subsequently, we perform the proposed Sinkhorn-type algorithm detailed in Algorithm 1, as well as the accelerated Sinkhorn-Newton-Sparse algorithm in Algorithm 2 (see Appendix C). As a benchmark, we provide numerical result for the APDAGD algorithm, whose detail is covered in Appendix D. We set the threshold parameters to be $t_I = t_E = \frac{1}{2}$, and it has been verified that the tested instances of equation 13 is feasible. We test the performance for Algorithm 1 and Algorithm 2. For Algorithm 2, we use $N_1 = 20$ Sinkhorn steps for initialization.

In Figure 2, we plot the performance of Algorithm 1 and Algorithm 2. One can see that the proposed Sinkhorn-type algorithm quickly converges to an approximately optimal solution that approximately satisfies the additional constraints. Algorithm 2 achieves similar performance under a vastly smaller number of iterations. In Figure 3, we show that both Algorithm 1 and Algorithm 2 can converge to $P_\eta^\star$ in the total variation (TV) distance. Furthermore, Algorithm 2 provides solutions with machine accuracy rapidly during the Newton stage. In contrast, while the initial performance of APDAGD is good, the convergence is relatively slow in later stages. This result is similar to the APDAGD

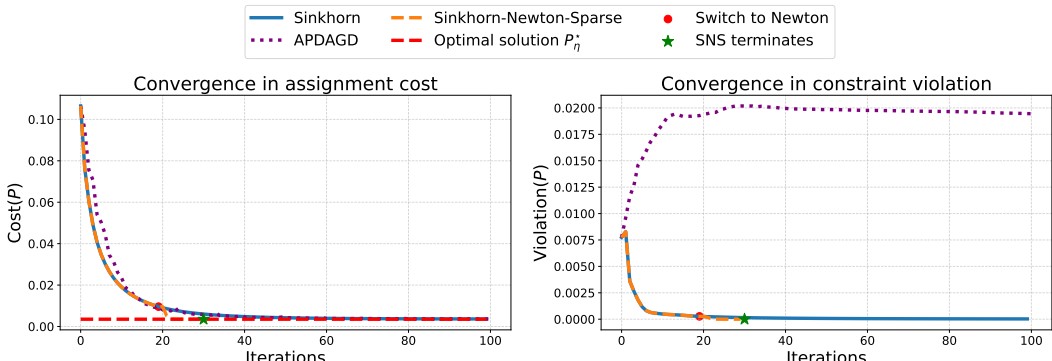

Figure 2: Random assignment problem. Plot of the proposed Sinkhorn-type algorithm in terms of assignment cost and constraint violation.

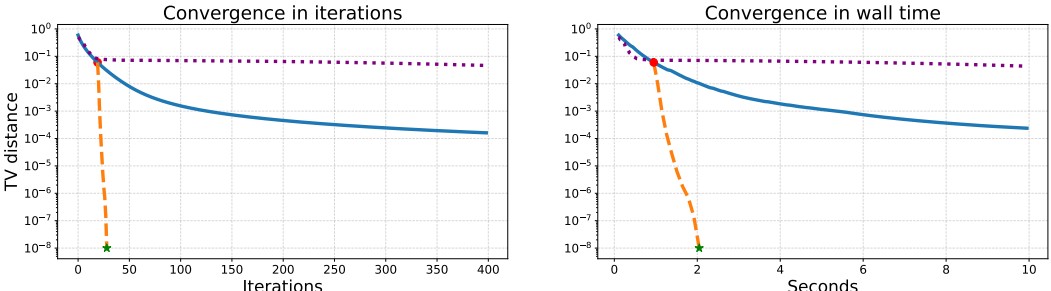

Figure 3: Random assignment problem. Plot of the proposed algorithms in terms of total variation (TV) distance to the entropically optimal solution $P_\eta^\star$.

algorithm in the unconstrained optimal transport setting, where likewise APDAGD could outperform Sinkhorn, but the benefit is largely limited to initial stages (Dvurechensky et al., 2018).

At last, to test the effect of random seed in the performance, we repeat the experiment 100 times. We plot the result in Figure 4, which shows that Algorithm 2 is quite robust in converging to $P_\eta^\star$. Further results are shown in Appendix B.

**Pareto front for geometric transport problem under Euclidean distance and Manhattan distance**   In the second numerical example, we consider the trade-off between two transport costs of a geometric nature. In this experiment, we run the task of Pareto front profiling between the Manhattan distance cost $c_1(x, y) = \|x - y\|_1$ and the Euclidean distance transport cost $c_2(x, y) = \|x - y\|_2^2$. As a byproduct, one obtains an interpolation between the Wasserstein $W_1$ transport plan under the $l_1$ distance and the $W_2$ transport plan under the $l_2$ distance. Let $C_1, C_2$ be the cost matrix associated with the Manhattan distance and the Euclidean distance (Villani et al., 2009). To do so, we consider the following optimization task:

$$\min_{P: P\mathbf{1}=r,\, P^\top \mathbf{1}=c} C_1 \cdot P,$$
$$\text{subject to} \quad C_2 \cdot P \leq t^2, \tag{14}$$

where $t$ is set so that the feasibility set is not empty. We let $t_{\min}$ be the $W_2$ distance between the source and target vectors, and let $t_{\max}$ be the Euclidean transport cost of a transport plan which minimizes Manhattan distance transport cost. We remark that $t_{\min}$ and $t_{\max}$ can be obtained through conventional OT algorithms. By tracing the value of $t$ in $[t_{\min}, t_{\max}]$ and solving equation 14 through entropic regularization, one can effectively obtain a Pareto front between Euclidean distance transport cost and Manhattan distance transport cost.

Similar to the previous case, Algorithm 2 is able to reach optimal solution $P_\eta^\star$ within machine accuracy. Thus, we run multiple instances of Algorithm 2 for $t \in [t_{\min}, t_{\max}]$ and $\eta = 10, 100, 1000$. Then, we plot the Pareto front formed by $P_\eta^\star$ for every choice of $\eta$. The true Pareto front can be

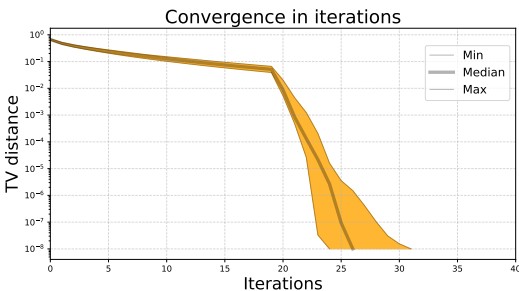

Figure 4: Random assignment problem. Aggregated performance of Algorithm 2 in terms of TV distance to $P_\eta^\star$ in random realizations of the random assignment problem across 100 random seeds.

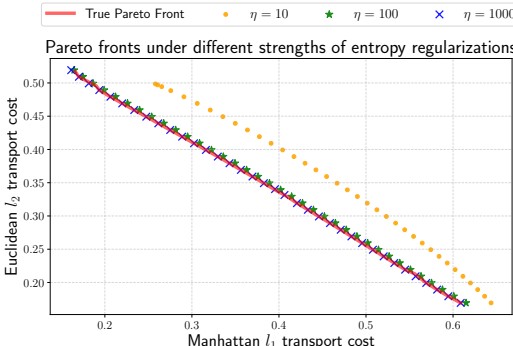

Figure 5: Pareto front profiling of Euclidean distance versus Manhattan distance.

obtained through running Algorithm 3 in Appendix C for $\eta_{\text{target}} = 8 \times 10^4$, and convergence is verified through checking the first order derivative reaches machine accuracy.

Similar to Cuturi (2013), we illustrate the procedure through optimal transport on the MNIST dataset. We pick two images, which are converted to a vector of intensities on the $28 \times 28$ pixel grid and normalized to sum to 1. The entry corresponding to the $(i_1, i_2)$-th pixel is conceptualized as the point $(i_1/28, i_2/28) \in \mathbb{R}^2$. In Figure 5, we see that the Pareto front formed by increasing $\eta$ indeed converges to the true Pareto front.

## 6    CONCLUSION

We introduce an entropic formulation of optimal transport with a combination of additional equality and inequality constraints. We propose a Sinkhorn-type algorithm that has a novel constraint dual variable update step. We provide preliminary results on the approximation error of the entropic formulation and the convergence of the Sinkhorn-type algorithm. A future direction is a more refined analysis on the convergence property of Algorithm 1 to the optimal entropic solution $P_\eta^\star$. Moreover, we provide a detailed discussion on improving the proposed Sinkhorn-type algorithm with acceleration techniques, especially Sinkhorn-Newton-Sparse and entropy regularization scheduling. The proposed work enables one to obtain approximately optimal solutions in more complicated OT instances efficiently. We contend that this work has the potential to be a vital tool in the field of optimal transport under constraint and in attracting the use of constrained optimal transport in machine learning. While the entropic barrier function for the constrained optimal transport has shown robust performance, the use of other barrier functions such as the log barrier might lend the formulation added flexibility and may enjoy good practical performance.

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

# A    APPLICATION TO RANKING PROBLEMS UNDER CONSTRAINTS

In this numerical example, we consider cost matrices associated with evaluation metrics in ranking problems (Liu et al., 2009; Manning, 2009). The problem of ranking can be naturally perceived as a transport problem where the transport matrix is analogous to the permutations of items (Mena et al., 2018). All linear additive ranking metrics, such as precision, recall, and discounted cumulative gain (DCG), are linear functions of the permutation matrix. For example, the DCG metric measures the quality of a ranking $\sigma : [n] \to [n]$. For a relevance vector $g \in \mathbb{R}^n$ and a discount vector $v \in \mathbb{R}^n$, a DCG score is defined by the following:

$$\mathrm{DCG}_{v,g}(\sigma) = \sum_{i=1}^{n} g_{\sigma(i)} v_i = (g v^\top) \cdot P$$

where $P$ is the permutation matrix associated with $\sigma$. Thus, optimizing the DCG metric corresponds to an OT instance through the relaxation of the permutation matrix within the Birkhoff polytope.

In practical problems such as e-commerce ranking, there are typically multiple relevance labels in the form of different attributes of an item. Motivated by this, we consider the following constrained OT problem:

$$
\begin{aligned}
\max_{P : P\mathbf{1}=\mathbf{1},\, P^\top \mathbf{1}=\mathbf{1}} \quad & (g_c v^\top) \cdot P, \\
\text{subject to} \quad & (g_I v^\top) \cdot P \geq t_I, \\
& (g_E v^\top) \cdot P = t_E,
\end{aligned}
\tag{15}
$$

where $g_c, g_I, g_E$ are three relevance vectors associated with the cost, the inequality constraint, and the equality constraint, respectively. In accordance with the information retrieval literature, we consider the discount vector $v$ with $v_i = \frac{1}{\log_2(i+1)}$. The entries of $g_c, g_I, g_E$ are i.i.d. entries simulated from the Rademacher distribution. In equation 15, we choose the two threshold variables as $t_I = \frac{1}{n} D_I \cdot \mathbf{1}_{n \times n}, t_E = \frac{1}{n} D_E \cdot \mathbf{1}_{n \times n}$, so that the feasibility set is guaranteed to be non-empty.

In accordance with equation 13, we consider a problem size of $n = 500$ and an entropy regularization of $\eta = \frac{1200}{500} = 2.4$. The performance is plotted in Figure 6 and Figure 7, which shows that both Algorithm 1 and Algorithm 2 can quickly converge to an approximately optimal solution that approximately satisfies the additional constraints.

# B    FURTHER DETAILS ON NUMERICAL EXPERIMENTS

**Statistics of performance under random assignment**    To fully justify the numerical finding in Section 5, we test the performance of the proposed algorithm under repeated sampling of random assignment problem parameters. In particular, the problem parameter $(C, D_I, D_E)$ is sampled by taking each entry to be i.i.d. random variables following $\mathrm{Unif}([0,1])$, and so different random seeds may lead to different performance. Thus, we use the problem setting in Section 5 and test the proposed algorithms under 100 random realizations. Figure 8 plots the performance of the algorithms in terms of total variation to the entropically optimal solution $P_\eta^\star$. One can see that the Sinkhorn-Newton-Sparse algorithm exhibits robust super-exponential convergence for all random assignment instances.

**Performance of random assignment under larger instances**    To show that the proposed algorithms are scalable to larger problem size, we use the experiment setting for Section 5 for random assignment problems. In particular, we take $n = 5000$, which is a much larger problem than the setting of $n = 500$ due to the $O(n^2)$ scaling. We use the same parameter setting as in Section 5. In Figure 9, we show that both the APDAGD algorithm continues to have relatively large constraint violation throughout the optimization procedure. In Figure 10, we show that both the Sinkhorn-type algorithm and the Sinkhorn-Newton-Sparse algorithm can converge to the entropically optimal solution, but the APDAGD algorithm stays quite far from the optimal solution.

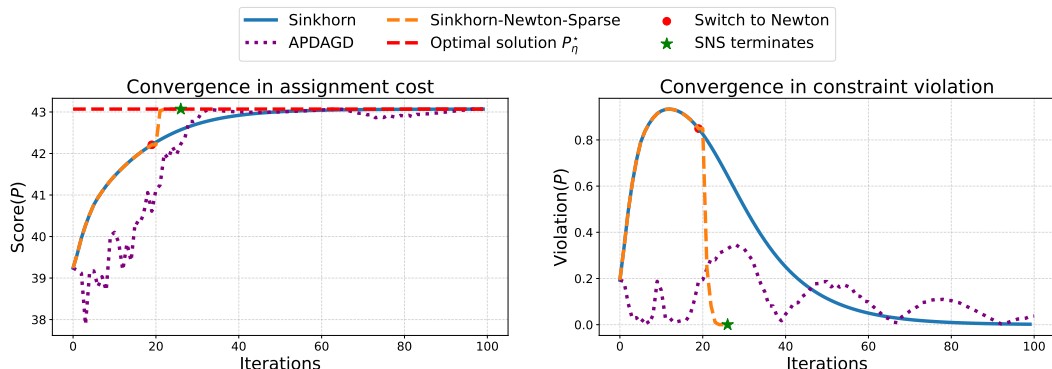

Figure 6: Ranking under constraints. Plot of the proposed Sinkhorn-type algorithm in terms of DCG score and constraint violation. Specifically, constraint violation is defined by $\mathrm{Violation}(P) = |\min(0, D_I \cdot \mathrm{Round}(P, \mathcal{U}_{r,c}))| + |D_E \cdot \mathrm{Round}(P, \mathcal{U}_{r,c})|$.

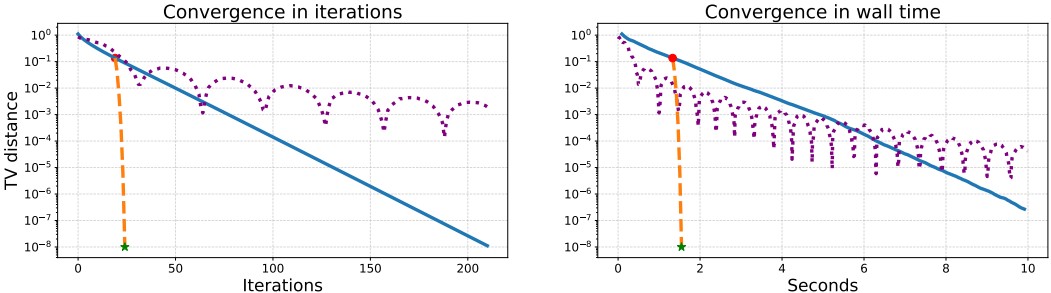

Figure 7: Ranking under constraints. Plot of the proposed algorithms in terms of TV distance to $P_\eta^\star$.

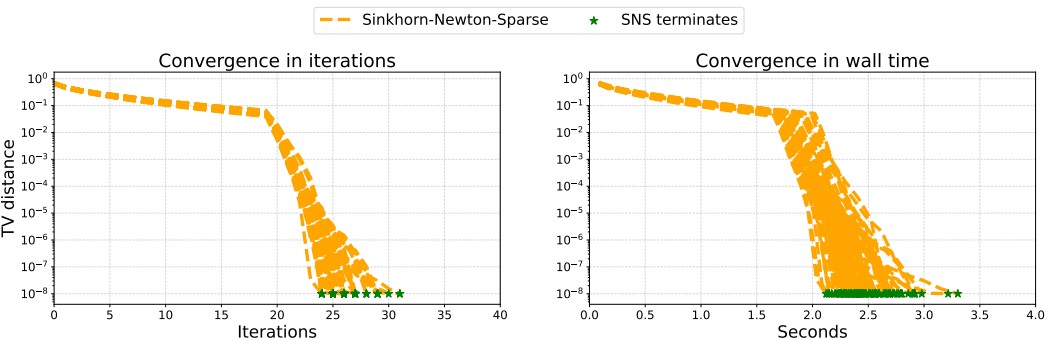

Figure 8: Plot of the proposed Sinkhorn-Newton-Sparse algorithm in terms of TV distance to $P_\eta^\star$ in 100 realizations.

## C  PRACTICAL IMPLEMENTATION OF ACCELERATED SINKHORN-TYPE ALGORITHM UNDER CONSTRAINT

In this section, we detail the procedure to combine Algorithm 1 with entropy regularization scheduling and Sinkhorn-Newton-Sparse (SNS). As the detail of entropic regularization scheduling is presented in Section 3, we shall give implementation details of SNS in the constrained case. Similar to the construction in Tang et al. (2024), for a matrix $M \in \mathbb{R}_{\geq 0}^{n \times n}$, we use $\mathrm{Sparsify}(M, \rho)$ to denote entry-wise truncation with a threshold $\rho$. For $\tilde{M} := \mathrm{Sparsify}(M, \rho)$, one has

$$\tilde{M}_{ij} = \begin{cases} M_{ij} & \text{if } M_{ij} \geq \rho, \\ 0 & \text{otherwise.} \end{cases}$$

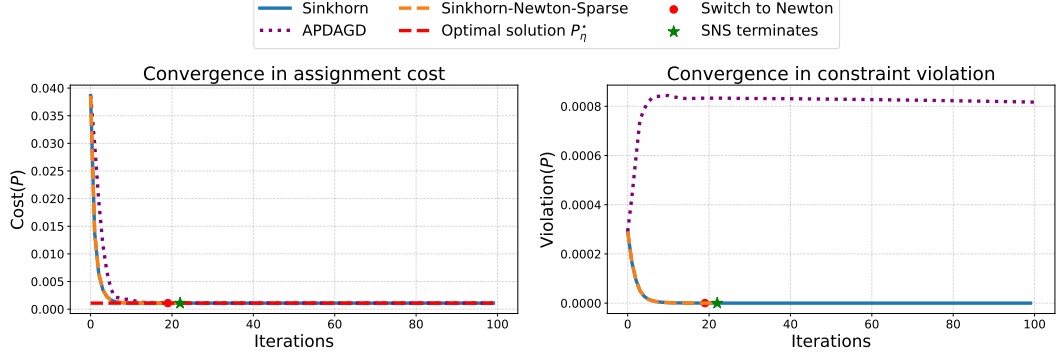

Figure 9: Random assignment problem for $n = 5000$. Plot of the proposed Sinkhorn-type algorithm in terms of the assignment cost and constraint violation.

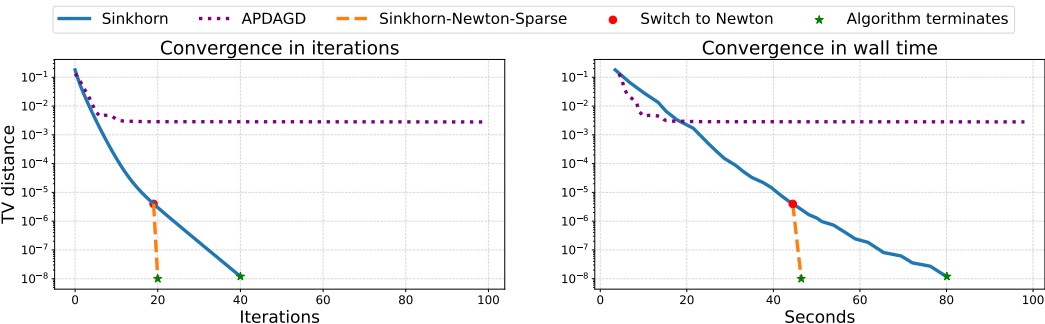

Figure 10: Random assignment problem for $n = 5000$. Plot of the proposed algorithms in terms of TV distance to $P_\eta^\star$.

Let $P$ be the intermediate transport matrix formed by the current dual variable $(x, y, a)$. One can write down the Hessian term $\nabla^2 f$ as follows:

$$\nabla^2 f(x, y, a) = -\eta \begin{bmatrix} \text{diag}(P\mathbf{1}) & P & \nabla_x \nabla_a f \\ P^\top & \text{diag}(P^\top \mathbf{1}) & \nabla_y \nabla_a f \\ \nabla_a \nabla_x f & \nabla_a \nabla_y f & \nabla_a^2 f \end{bmatrix}. \tag{16}$$

As we assume $K + L = O(1)$ in this work, the terms such as $\nabla_{xy}\nabla_a f, \nabla_a^2 f$ can be kept without posing significant challenges to the Newton step, which allows us to use the approximation below:

$$\nabla^2 f(x, y, a) \approx H = -\eta \begin{bmatrix} \text{diag}(P\mathbf{1}) & \text{Sparisfy}(P, \rho) & \nabla_x \nabla_a f \\ \text{Sparisfy}(P^\top, \rho) & \text{diag}(P^\top \mathbf{1}) & \nabla_y \nabla_a f \\ \nabla_a \nabla_x f & \nabla_a \nabla_y f & \nabla_a^2 f \end{bmatrix}. \tag{17}$$

The value of $\rho$ is a tunable parameter, and one sets $\rho$ so that $\text{Sparisfy}(P, \rho)$ contains only $O(n)$ nonzero entries. As a result, applying $H^{-1}$ to a vector can be done with a $O((n+K+L)^2) = O(n^2)$ complexity through the conjugate-gradient algorithm (Golub and Van Loan, 2013). Additionally, to ensure stability in the degenerate direction $(\delta x, \delta y, \delta a) = v := (\mathbf{1}_n, -\mathbf{1}_n, \mathbf{0}_{K+L})$, we in practice use the following modified version of Lyapunov function:

$$\tilde{f}(x, y, a) = f(x, y, a) - \frac{1}{2}(\sum_i x_i - \sum_j y_j)^2.$$

Same as Tang et al. (2024), one can see that the maximizer of $\tilde{f}$ is also a maximizer of $f$. The final Sinkhorn-Newton-Sparse algorithm used is in Algorithm 2, where we include a Sinkhorn stage which uses Algorithm 1 as initialization, and a subsequent Newton stage which uses sparse Newton iteration to accelerate convergence.

The combination of entropy regularization scheduling and SNS is described in Algorithm 3. We remark that the iteration count $N_{1,i}, N_{2,i}$ within Algorithm 3 are typically set to be much smaller than $N_1, N_2$ in Algorithm 2, which is possible because the optimization task at an entropy regularization parameter $\eta_i$ is initialized by the dual variables obtained for regularization parameter $\eta_{i-1} = \eta_i/2$.

As a special case, Algorithm 3 also provides substantial acceleration to the unconstrained entropic OT problem (Cuturi, 2013).

## D  ACCELERATED FIRST-ORDER METHOD FOR CONSTRAINED OT

In this section, we detail the implementation for the adaptive primal-dual accelerated gradient descent (APDAGD) method introduced in Dvurechensky et al. (2018). In particular, the APDAGD algorithm already generalizes to the setting of constrained optimal transport by simply using the primal-dual form in equation 6. Therefore, we will only describe the algorithm detail of APDAGD. A future direction is to analyze the APDAGD algorithm for constrained optimal transport and see if one can likewise obtain complexity guarantee similar to Lin et al. (2019). For simplicity, we do not use the adaptive termination condition, and instead we simply run the APDAGD condition for a fixed number of iterations.

**Algorithm summary**  We summarize the procedure in Algorithm 4, which implements the APDAGD algorithm using the notation of this work. The term $f$ is the Lyapunov dual potential defined in equation 7.

## E  PARTIAL OPTIMAL TRANSPORT UNDER ENTROPIC REGULARIZATION

Partial optimal transport (POT) considers the following linear programming problem:

$$\min_{P: P\mathbf{1} \leq r, P^\top \mathbf{1} \leq c, \mathbf{1}^\top P\mathbf{1} = t, P \geq 0} C \cdot P, \tag{18}$$

which has found great application in machine learning and general engineering applications (Sarlin et al., 2020; Chapel et al., 2020; Bonneel and Coeurjolly, 2019; Liu et al., 2020; Kawano et al.,

---

**Algorithm 2** Sinkhorn-Newton-Sparse (SNS) for OT under linear constraint

---

**Require:** $\tilde{f}, x_{\mathrm{init}} \in \mathbb{R}^n, y_{\mathrm{init}} \in \mathbb{R}^n, a_{\mathrm{init}} \in \mathbb{R}^{K+L}, N_1, N_2, \rho, i = 0$

1: # Sinkhorn stage

2: $v \leftarrow \begin{bmatrix} \mathbf{1}_n \\ -\mathbf{1}_n \\ \mathbf{0}_{K+L} \end{bmatrix}$      $\triangleright$ Initialize degenerate direction

3: $(x, y, a) \leftarrow (x_{\mathrm{init}}, y_{\mathrm{init}}, a_{\mathrm{init}})$      $\triangleright$ Initialize dual variable

4: **while** $i < N$ **do**

5:     $i \leftarrow i + 1$

6:     # Row&Column scaling step

7:     $P \leftarrow \exp\left(\eta(-C + \sum_m a_m D_m + x\mathbf{1}^\top + \mathbf{1}y^\top) - 1\right)$

8:     $x \leftarrow x + \left(\log(r) - \log(P\mathbf{1})\right)/\eta$

9:     $P \leftarrow \exp\left(\eta(-C + \sum_m a_m D_m + x\mathbf{1}^\top + \mathbf{1}y^\top) - 1\right)$

10:     $y \leftarrow y + \left(\log(c) - \log(P^\top\mathbf{1})\right)/\eta$

11:     # Constraint dual update step

12:     $a, t \leftarrow \arg\max_{\tilde{a}, \tilde{t}} f(x + \tilde{t}\mathbf{1}, y, \tilde{a})$

13:     $x \leftarrow x + t\mathbf{1}$

14: **end while**

15: # Newton stage

16: $z \leftarrow \mathrm{Proj}_{v^\perp}((x, y, a))$      $\triangleright$ Project into non-degenerate direction of $f$

17: **while** $i < N_1 + N_2$ **do**

18:     $P \leftarrow \exp\left(\eta(-C + \sum_m a_m D_m + x\mathbf{1}^\top + \mathbf{1}y^\top) - 1\right)$

19:     $H \leftarrow -\eta \begin{bmatrix} \mathrm{diag}(P\mathbf{1}) & \mathrm{Sparsify}(P, \rho) & \nabla_x\nabla_a f \\ \mathrm{Sparsify}(P^\top, \rho) & \mathrm{diag}(P^\top\mathbf{1}) & \nabla_y\nabla_a f \\ \nabla_a\nabla_x f & \nabla_a\nabla_y f & \nabla_a^2 f \end{bmatrix}$    $\triangleright$ Sparse approximation of $\nabla^2 f$

    with threshold $\rho$.

20:     $H \leftarrow H - vv^\top$      $\triangleright$ Add regularization term corresponding to $\tilde{f}$.

21:     $\Delta z \leftarrow \mathrm{Conjugate\_Gradient}(H, -\nabla\tilde{f}(z))$      $\triangleright$ Solve sparse linear system

22:     $\alpha \leftarrow \mathrm{Line\_search}(\tilde{f}, z, \Delta z)$      $\triangleright$ Line search for step size

23:     $z \leftarrow z + \alpha\Delta z$

24:     $i \leftarrow i + 1$

25: **end while**

26: Output dual variables $(x, y, a) \leftarrow z$.

---

**Algorithm 3** Sinkhorn-Newton-Sparse with entropy regularization scheduling for OT under linear constraint

---

**Require:** $x_{\mathrm{init}} \in \mathbb{R}^n, y_{\mathrm{init}} \in \mathbb{R}^n, a_{\mathrm{init}} \in \mathbb{R}^{K+L}, \rho$

**Require:** $\eta_{\mathrm{target}}, N_\eta = \lceil\log_2(\eta_{\mathrm{target}})\rceil, (N_{1,i})_{i=1}^{N_\eta}, (N_{2,i})_{i=1}^{N_\eta}, i = 1$

1: $(x, y, a) \leftarrow (x_{\mathrm{init}}, y_{\mathrm{init}}, a_{\mathrm{init}})$      $\triangleright$ Initialize dual variable

2: $\eta = 1$      $\triangleright$ Initialize entropy regularization

3: **while** $i \leq N_\eta$ **do**

4:     Run Algorithm 2 with entropy regularization set to $\eta$ and initialized dual variables set to $(x, y, a)$, and $N_1, N_2$ set to $N_{1,i}, N_{2,i}$.

5:     Save the output of previous step to $(x, y, a)$.

6:     $i \leftarrow i + 1$

7:     $\eta = \min(2\eta, \eta_{\mathrm{target}})$      $\triangleright$ Double entropy regularization term

8: **end while**

9: Run the Newton stage of Algorithm 2 at $\eta = \eta_{\mathrm{target}}$ until the solution reaches convergence.

10: Output dual variables $(x, y, a)$.

---

---

**Algorithm 4** Adaptive primal-dual accelerated gradient descent algorithm (APDAGD)

---

**Require:** $f, N, k = 0, z_0 = \zeta_0 = \lambda_0 = 0_{2n+m}$
 1: $\alpha_0 \leftarrow 0, \beta_0 \leftarrow 0, L_0 = 1,$
 2: **while** $k < N$ **do**
 3:     $M_k = L_k/2$
 4:     **while** True **do**
 5:         $M_k = 2M_k$
 6:         $\alpha_{k+1} = \frac{1+\sqrt{1+4M_k\beta_k}}{2M_k}$
 7:         $\beta_{k+1} = \beta_k + \alpha_{k+1}$
 8:         $\tau_k = \frac{\alpha_{k+1}}{\beta_{k+1}}$
 9:         $\lambda_{k+1} \leftarrow \tau_k\zeta_k + (1 - \tau_k)z_k$
 10:        $\zeta_{k+1} \leftarrow \zeta_k + \alpha_{k+1}\nabla f(\lambda_{k+1})$
 11:        $z_{k+1} \leftarrow \tau_k\zeta_{k+1} + (1 - \tau_k)z_k$
 12:        **if** $f(z_{k+1}) \geq f(\lambda_{k+1}) + \langle \nabla f(\lambda_{k+1}), z_{k+1} - \lambda_{k+1} \rangle - \frac{M_k}{2}\|z_{k+1} - \lambda_{k+1}\|_2^2$ **then**
 13:            Break
 14:        **end if**
 15:    **end while**
 16:    $L_{k+1} \leftarrow M_k/2k \leftarrow k + 1$
 17: **end while**
 18: Output dual variables $(x, y, a) \leftarrow z_{N-1}$.

---

2022; Wang et al., 2022; Nietert et al., 2023). In particular, the row and column inequality constraint makes the POT formulation different from the constrained optimal transport setting formulated in equation 4.

**Variational formulation of entropically regularized POT**   Similar to the main setting of constrained optimal transport, the POT task in equation 18 also admits an entropic regularization following the entropic LP formulation (Fang, 1992), and one writes down the primal formulation as follows:

$$\min_{P,p,q:P\mathbf{1}+p=r, P^\top\mathbf{1}+q=c, \mathbf{1}^\top P\mathbf{1}=t} C \cdot P + \frac{1}{\eta}H(P, p, q), \tag{19}$$

where the entropy term is defined by

$$H(P, p, q) = \sum_{ij} p_{ij}\log(p_{ij}) + \sum_i p_i\log p_i + \sum_j q_j\log q_j.$$

We designate $x, y \in \mathbb{R}^n$ as the symbol for dual variables corresponding to the row and column constraint, and we designate $w \in \mathbb{R}$ as the symbol for the dual variable for the total sum constraint. By performing similar derivation with Lagrangian dual variable and minimax theorem as in Section 2, one obtains the following dual objective for POT:

$$
\begin{aligned}
f_{\mathrm{POT}}(x, y, w) = &-\frac{1}{\eta}\sum_{ij}\exp\left(\eta(-C_{ij} + w + x_i + y_j) - 1\right) + tw \\
&+ \sum_i x_i r_i + \sum_j y_j c_j - \frac{1}{\eta}\sum_i\exp(\eta x_i - 1) - \frac{1}{\eta}\sum_j\exp(\eta y_j - 1),
\end{aligned}
\tag{20}
$$

and we remark that equation 20 is equivalent to the dual formulation of the entropically regularized POT as in Nguyen et al. (2024). In particular, let $P = [\exp(\eta(-C_{ij} + w + x_i + y_j) - 1)]_{ij}$ denote the intermediate transport plan variable formed by the dual variable $(x, y, w)$, and likewise we use $p = [\exp(\eta x_i - 1)]_i$, $q = [\exp(\eta y_j - 1)]_j$ to denote the respective row and column slack variables formed by the dual variable. By simple calculation, one has

$$\partial_x f_{\mathrm{POT}} = r - P\mathbf{1} - p, \quad \partial_y f_{\mathrm{POT}} = c - P^\top\mathbf{1} - q, \quad \partial_w f_{\mathrm{POT}} = t - \mathbf{1}^\top P\mathbf{1},$$

which in particular verifies that stationarity of $f_{\mathrm{POT}}$ implies the $P$ variable gives a feasible POT solution.

**Sparse Newton iterations for POT**  In this section, we propose to use sparse Newton iterations (Tang et al., 2024) to form an accelerated algorithm for the POT dual objective. In particular, one can calculate the Hessian and obtain

$$\nabla^2 f_{\text{POT}} = -\eta \begin{bmatrix} \text{diag}(P\mathbf{1} + p) & P & P\mathbf{1} \\ P^\top & \text{diag}(P^\top\mathbf{1} + q) & P^\top\mathbf{1} \\ (P\mathbf{1})^\top & (P^\top\mathbf{1})^\top & \mathbf{1}^\top P\mathbf{1} \end{bmatrix},$$

for which one can see that $\nabla^2 f_{\text{POT}}$ admits a sparse approximation as long as $P$ admits a sparse approximation. Assuming uniqueness, the optimal solution $P^\star$ to equation 18 has only $O(n)$ entries due to the fundamental theorem of linear programming. Moreover, the entropically optimal regularized solution $P^\star_\eta$ to equation 19 is exponentially close to $P^\star$ following the analysis in Weed (2018). Therefore, it is reasonable to assume that $P$ admits a sparse approximation and one can apply a sparse approximation of $\nabla^2 f_{\text{POT}}$ for second-order acceleration. Notably, the application of sparse Newton iteration for the POT task calls for the design of a warm initialization procedure, and one can directly apply the APDAGD algorithm in Nguyen et al. (2024). As the sparse approximation only occurs for the preconditioner of the optimization procedure, the sparsification error doesn't affect convergence to ground truth as long as the error is mild.

We remark that one can also use entropy regularization scheduling (Chen et al., 2023) to gradually increase the entropy regularization parameter to $\eta$, which would also achieve warm initialization.

**Algorithm summary**  To summarize, we propose a two-stage algorithm for the entropically regularized POT task in equation 19. The two-stage approach is such that the first stage uses the APDAGD algorithm for fixed number of iterations, and the second stage uses sparse Newton iterations to reach convergence. The algorithm block is in Algorithm 5. We remark that the APDAGD procedure for POT can be found in Nguyen et al. (2024), and one can also use APDAGD by running Algorithm 4 in Appendix D with $f_{\text{POT}}$ in place of $f$.

---

**Algorithm 5** APDAGD followed by sparse Newton iteration for POT

---

**Require:** $f_{\text{POT}}, N_1, N_2, \rho, i = 0$
1: # APDAGD stage
2: Run $N_1$ iterations of APDAGD and output the dual variable $(x, y, w)$ as initialization
3: # Newton stage
4: **while** $i < N_2$ **do**
5:     $(x, y, w) \leftarrow z$
6:     $P \leftarrow \exp\left(\eta(-C + x\mathbf{1}^\top + \mathbf{1}y^\top + w) - 1\right)$
7:     $p \leftarrow \exp\left(\eta x - 1\right), \quad q \leftarrow \exp\left(\eta y - 1\right)$
8:     $H \leftarrow -\eta \begin{bmatrix} \text{diag}(P\mathbf{1} + p) & \text{Sparisfy}(P, \rho) & P\mathbf{1} \\ \text{Sparisfy}(P^\top, \rho) & \text{diag}(P^\top\mathbf{1} + q) & P^\top\mathbf{1} \\ (P\mathbf{1})^\top & (P^\top\mathbf{1})^\top & \mathbf{1}^\top P\mathbf{1} \end{bmatrix}$    ▷ Sparse Hessian approximation
    with threshold $\rho$.
9:     $\Delta z \leftarrow \text{Conjugate\_Gradient}(H, -\nabla f_{\text{POT}}(z))$          ▷ Solve sparse linear system
10:    $\alpha \leftarrow \text{Line\_search}(\tilde{f}, z, \Delta z)$                ▷ Line search for step size
11:    $z \leftarrow z + \alpha\Delta z$
12:    $i \leftarrow i + 1$
13: **end while**
14: Output dual variables $(x, y, w) \leftarrow z$.

---

**Numerical experiment for POT**  We test the performance of the proposed algorithm. As a comparison, we use the APDAGD algorithm in Nguyen et al. (2024) as a benchmark. The metric used is the optimality gap of the dual objective $f_{\text{POT}}$. For a fair comparison, we choose the same entropic parameter for APDAGD and for the sparse Newton algorithm, and we run the APDAGD algorithm for a fixed number of iterations.

In the numerical test, we consider the random assignment problem (Mézard and Parisi, 1987; Steele, 1997; Aldous, 2001). The cost matrix $C = [c_{ij}]_{ij=1}^n \in \mathbb{R}^{n \times n}$ with $n = 500$ is generated by $c_{ij} \sim \text{Unif}([0, 1])$, and we take $t = 1/2$ and let $r = c = \frac{0.51}{n}\mathbf{1}$. We use $N_1 = 20$ for Algorithm

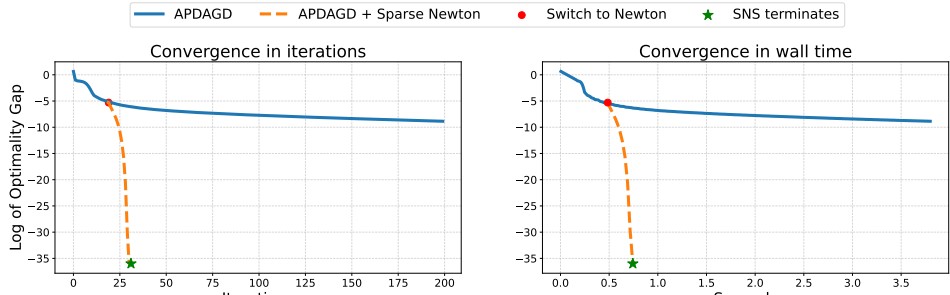

Figure 11: Performance comparison between Algorithm 5 and the APDAGD algorithm.

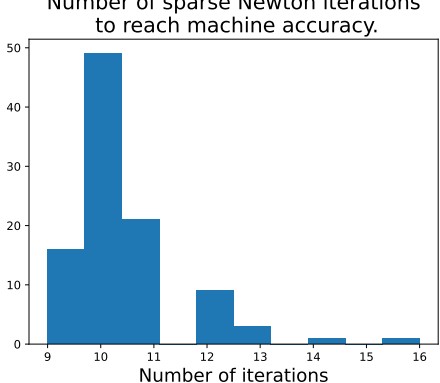

Figure 12: Histogram for the number of sparse Newton iterations to reach machine accuracy for 100 independent sampling of random linear assignment tasks.

5 and we sparsify the Hessian so that one keeps only $2/n$ fraction of the entries. As seen in Figure 11, the sparse Newton iteration is able to converge to the optimal solution after a few iterations. As the cost matrix is generated randomly, we repeat the same experiment 100 times, and Figure 12 shows that the super-exponential convergence is robust across iterations. In particular, the constraint violation for SNS is with machine accuracy after a few steps of iterations, which is why one does not necessarily need to use the rounding step in Nguyen et al. (2024), but larger values of $\eta$ might necessitate the use of projection to ensure constraint satisfaction.

## F    PROOF OF THEOREM 1

**Definition 1.** Let $\mathcal{S}$ be the constraint set defined in equation 4. Define $\mathcal{P}$ as the polyhedron formed by the transport matrix, i.e.

$$\mathcal{P} := \{P \mid P\mathbf{1} = r, P^\top \mathbf{1} = c, P \geq 0, P \in \mathcal{S}\}.$$

The symbol $\mathcal{V}$ denotes the set of vertices of $\mathcal{P}$. The symbol $\mathcal{O}$ stands for the set of optimal vertex solutions, i.e.

$$\mathcal{O} := \arg\min_{P \in \mathcal{V}} C \cdot P. \tag{21}$$

The symbol $\Delta$ denotes the vertex optimality gap

$$\Delta = \min_{Q \in \mathcal{V} - \mathcal{O}} Q \cdot C - \min_{P \in \mathcal{O}} P \cdot C.$$

We can now finish the proof.

*Proof.* This convergence result is mainly due to the application of Corollary 9 in Weed (2018) to this case. We define another polyhedron $\mathcal{Q}$ as follows:

$$\mathcal{Q} := \{(P, s) \mid P \in \mathcal{P}, \forall k \in [K], s_k = D_k \cdot P\}.$$

Let $P_\eta^\star$ be as defined in the statement, and for $k = 1, \ldots, K$ we define $s_{\eta;k}^\star = D_k \cdot P_\eta^\star$. We use $R_1$ and $R_H$ to denote the $l_1$ and entropic radius of $\mathcal{Q}$ in the sense defined in Weed (2018). It is easy to see that for $R_1$ one has

$$1 \leq R_1 = 1 + \max_{P \in \mathcal{P}} \sum_{k=1}^{K} P \cdot D_k \leq 1 + K,$$

where the second inequality uses Holder's inequality and the assumption that $\|D_k\|_\infty \leq 1$

For $R_H$, one similarly has

$$R_H = \max_{(s,P),(s',P') \in \mathcal{Q}} \sum_{ij} \left( p_{ij} \log(p_{ij}) - p'_{ij} \log(p'_{ij}) \right) + \sum_k \left( s_k \log(s_k) - s'_k \log(s'_k) \right)$$

$$\leq \left( \max_{P,P' \in \mathcal{P}} \sum_{ij} \left( p_{ij} \log(p_{ij}) - p'_{ij} \log(p'_{ij}) \right) \right) + \left( \max_{P,P' \in \mathcal{P}} \sum_k (P \cdot D_k) \log(P \cdot D_k) - (P' \cdot D_k) \log(P' \cdot D_k) \right)$$

$$\leq \log(n^2) + K/e,$$

where the second equality holds because $H(P) \in [0, \log(n^2)]$ and $P \cdot D_k \log(P \cdot D_k) \in [-1/e, 0]$.

For $\eta \geq \frac{(K+1)(1+\ln(4n^2(K+1)))}{\Delta} > \frac{R_1 + R_H}{\Delta}$, one has

$$\|P^\star - P_\eta^\star\|_1 \leq \|(P^\star, s^\star) - (P_\eta^\star, s_\eta^\star)\|_1$$

$$\leq 2R_1 \exp\left( -\eta \frac{\Delta}{R_1} + 1 + \frac{R_H}{R_1} \right)$$

$$= 2R_1 \exp\left( \frac{R_H - \eta\Delta}{R_1} + 1 \right)$$

$$\leq 2(K+1) \exp\left( \frac{R_H - \eta\Delta}{K+1} + 1 \right)$$

$$= 2(K+1) \exp\left( \frac{2\log(n) + K/e - \eta\Delta}{K+1} + 1 \right)$$

$$\leq 8n^{\frac{2}{K+1}}(K+1) \exp\left( -\eta \frac{\Delta}{K+1} \right),$$

where the third inequality is because $R_H - \eta\Delta \leq 0$, and the last inequality holds because $\exp(\frac{K/e}{K+1} + 1) \leq \exp(1 + 1/e) \leq 4$. $\square$

# G   CHARACTERIZATION OF LYAPUNOV FUNCTION UNDER SINKHORN STEPS

Before proving Theorem 2, we first prove the following theorem which characterizes the improvement of the Lyapunov function from one step of the proposed algorithm:

**Theorem 3.** *Let $(x, y, a)$ be the current dual variable which has undergone at least one update step, and let $P$ be the associated transport matrix, i.e.,*

$$P = \exp\left( \eta(-C + \sum_m a_m D_m + x\mathbf{1}^\top + \mathbf{1}y^\top) - 1 \right).$$

*For the $x$ update step satisfying $x' = \arg\max_{\tilde{x}} f(\tilde{x}, y, a)$, one has*

$$f(x', y, a) - f(x, y, a) = \mathrm{KL}\left( r \| P\mathbf{1} \right), \tag{22}$$

*where $\mathrm{KL}(\cdot\|\cdot)$ stands for the Kullback-Leibler divergence.*

*For the $y$ update step satisfying $y' = \arg\max_{\tilde{y}} f(x, \tilde{y}, a)$, one has*

$$f(x, y', a) - f(x, y, a) = \mathrm{KL}\left( c \| P^\top\mathbf{1} \right). \tag{23}$$

*Consider the $a$ update step satisfying $(a', t') = \arg\max_{\tilde{a}, \tilde{t}} f(x + \tilde{t}\mathbf{1}, y, \tilde{a})$. For $m = 1, \ldots, K + L$, define $d_m$ as the optimality condition corresponding to $a_m$, i.e.*

$$
\begin{aligned}
d_k &:= \exp(-\eta a_k - 1) - P \cdot D_k, \quad k = 1, \ldots, K \\
d_{l+K} &:= -P \cdot D_{l+K}, \qquad\qquad\qquad l = 1, \ldots, L.
\end{aligned}
\tag{24}
$$

*Then, for $c_d := \max_{m \in [K+L]} \|D_m\|_\infty$, one has*

$$
\begin{aligned}
&f(x + t'\mathbf{1}, y, a') - f(x, y, a) \\
&\geq \sum_{k=1}^{K} |d_k| \min\left( \frac{1}{8\eta}, \frac{|d_k|}{8\eta c_d + 4\eta(K + L)c_d^2} \right) \\
&+ \sum_{l=1}^{L} \frac{d_{l+K}^2}{2\eta(K + L)c_d^2}.
\end{aligned}
\tag{25}
$$

For legibility, in what follows, we introduce the symbols $\hat{x}, \hat{y}, \hat{a}$ used exclusively for dummy variables.

*Proof.* For the reader's convenience, we list the explicit form of the Lyapunov function $f$ here:

$$
f(\hat{x}, \hat{y}, \hat{a}) = -\frac{1}{\eta} \sum_{ij} \exp\left( \eta(-C_{ij} + \sum_{m=1}^{L+K} \hat{a}_m (D_m)_{ij} + \hat{x}_i + \hat{y}_j) - 1 \right) + \sum_i \hat{x}_i r_i + \sum_j \hat{y}_j c_j - \frac{1}{\eta} \sum_{k=1}^{K} \exp(-\eta \hat{a}_k - 1).
$$

First we prove $\sum_{ij} P_{ij} = 1$. By assumption, the dual variable $(x, y, a)$ has gone through at least one update step. If the $x$ update step has been last performed, then one has $\nabla_x f = 0$, which implies $\sum_{ij} P_{ij} = \sum_i r_i = 1$. If the $y$ update step has been last performed, then one likewise has $\nabla_y f = 0$ and $\sum_{ij} P_{ij} = \sum_j c_j = 1$. If the $a$ update step has been performed, then the optimality in the $t$ variable as shown in equation 9 implies $\sum_{ij} P_{ij} = 1$. Thus, one has $\sum_{ij} P_{ij} = 1$ in all the three possible cases, as claimed.

The proof for equation 22 and equation 23 then largely follows from Lemma 2 of (Altschuler et al., 2017). Suppose that an $x$ update step is performed and $P'$ is the matrix formed by the dual variable $(x', y, a)$. Due to the optimality of the $x'$ variable, one has $\sum_{ij} P'_{ij} = 1$. Thus one has

$$
f(x', y, a) - f(x, y, a) = \frac{1}{\eta}\left( \sum_{ij} P_{ij} - \sum_{ij} P'_{ij} \right) + \sum_i r_i(x'_i - x_i) = \sum_i r_i(x'_i - x_i) = \mathrm{KL}\left( r \| P\mathbf{1} \right),
$$

where the last equality is due to $r_i(x'_i - x_i) = r_i \log \frac{r_i}{(P\mathbf{1})_i}$. The proof for equation 23 follows likewise:

$$
f(x, y', a) - f(x, y, a) = \frac{1}{\eta}\left( \sum_{ij} P_{ij} - \sum_{ij} P'_{ij} \right) + \sum_j c_j(y'_j - y_j) = \sum_j c_j(y'_j - y_j) = \mathrm{KL}\left( c \| P^\top \mathbf{1} \right).
$$

We proceed with the proof for equation 25. First, we introduce an augmented Lyapunov function by maximizing over the $t$ variable, which provides a smoother optimization landscape for subsequent analysis. Define the augmented Lyapunov function $f_{\mathrm{aug}}$ as

$$
f_{\mathrm{aug}}(\hat{x}, \hat{y}, \hat{a}) := \max_{\tilde{t}} f(\hat{x} + \tilde{t}\mathbf{1}, \hat{y}, \hat{a}).
$$

As $\sum_{ij} P_{ij} = 1$ implies optimality over the $t$ variable by equation 9, one has $f(x, y, a) = f_{\mathrm{aug}}(x, y, a)$ for the dual variable $(x, y, a)$. After the augmented $a$ update step, one has by definition that $f(x + t'\mathbf{1}, y, a') = f_{\mathrm{aug}}(x, y, a')$. Thus, one can simplify the left-hand side of equation 25 by the following equation:

$$
f(x + t'\mathbf{1}, y, a') - f(x, y, a) = f_{\mathrm{aug}}(x, y, a') - f_{\mathrm{aug}}(x, y, a) = \max_{\tilde{a}} f_{\mathrm{aug}}(x, y, \tilde{a}) - f_{\mathrm{aug}}(x, y, a).
$$

We then derive the formula for $f_{\mathrm{aug}}$. By direct calculation, one has

$$
f(\hat{x} + t\mathbf{1}, \hat{y}, \hat{a})
$$

$$
= -\frac{1}{\eta} \sum_{ij} \exp\left( \eta(-C_{ij} + \sum_{m=1}^{L+K} \hat{a}_m (D_m)_{ij} + \hat{x}_i + \hat{y}_j + t) - 1 \right) + \sum_i \hat{x}_i r_i + \sum_j \hat{y}_j c_j + t - \frac{1}{\eta} \sum_{k=1}^{K} \exp(-\eta \hat{a}_k - 1).
$$

Let $t^\star = \arg\max_{\tilde{t}} f(\hat{x} + \tilde{t}\mathbf{1}, \hat{y}, \hat{a})$. Under the optimality condition $\partial_t f(\hat{x} + t^\star\mathbf{1}, \hat{y}, \hat{a}) = 0$, one has

$$\sum_{ij} \exp\left(\eta(-C_{ij} + \sum_{m=1}^{L+K} \hat{a}_m (D_m)_{ij} + \hat{x}_i + \hat{y}_j + t^\star) - 1\right) = 1.$$

Taking log over the above equation, one has the following result for $t^\star$:

$$1 - \eta t^\star = \mathrm{LSE}\left(\eta(-C + \sum_m \hat{a}_m D_m + \hat{x}\mathbf{1}^\top + \mathbf{1}\hat{y}^\top)\right), \tag{26}$$

where for a matrix $M$, the function LSE defines the log-sum-exponential function $\mathrm{LSE}(M) = \log\left(\sum_{ij} \exp(m_{ij})\right)$. Thus one has

$$\begin{aligned}
f_{\mathrm{aug}}(\hat{x}, \hat{y}, \hat{a}) =& f(\hat{x} + t^\star\mathbf{1}, \hat{y}, \hat{a}) \\
=& -\frac{1}{\eta} + \sum_i \hat{x}_i r_i + \sum_j \hat{y}_j c_j + t^\star - \frac{1}{\eta}\sum_{k=1}^K \exp(-\eta\hat{a}_k - 1) \\
=& -\frac{1}{\eta}\mathrm{LSE}\left(\eta(-C + \sum_m \hat{a}_m D_m + \hat{x}\mathbf{1}^\top + \mathbf{1}\hat{y}^\top)\right) + \sum_i \hat{x}_i r_i + \sum_j \hat{y}_j c_j - \frac{1}{\eta}\sum_{k=1}^K \exp(-\eta\hat{a}_k - 1).
\end{aligned}$$

The rest of the proof follows from a perturbational argument on $f_{\mathrm{aug}}$ around the point $(x, y, a)$, and thus we provide a formula for the derivatives of $f_{\mathrm{aug}}$. Let $\hat{P} = \exp\left(\eta(-C + \sum_m \hat{a}_m D_m + \hat{x}\mathbf{1}^\top + \mathbf{1}\hat{y}^\top) - 1\right)$ be the transport plan formed by the dual variable $(\hat{x}, \hat{y}, \hat{a})$. For $k = 1, \ldots, K$, one has

$$\partial_{a_k} f_{\mathrm{aug}}(\hat{x}, \hat{y}, \hat{a}) = \exp(-\eta\hat{a}_k - 1) - D_k \cdot \frac{\hat{P}}{\sum_{ij}\hat{P}_{ij}},$$

and likewise for $l = 1, \ldots, L$ one has

$$\partial_{a_{l+K}} f_{\mathrm{aug}}(\hat{x}, \hat{y}, \hat{a}) = -D_{l+K} \cdot \frac{\hat{P}}{\sum_{ij}\hat{P}_{ij}}.$$

In terms of the second-order information, by direct calculation, one has

$$\partial_{a_m}\partial_{a_{m'}} f_{\mathrm{aug}}(\hat{x}, \hat{y}, \hat{a}) = -\eta(D_m \odot D_{m'}) \cdot \frac{\hat{P}}{\sum_{ij}\hat{P}_{ij}} + \eta\frac{\left(\hat{P} \cdot D_m\right)\left(\hat{P} \cdot D_{m'}\right)}{\left(\sum_{ij}\hat{P}_{ij}\right)^2} + s_{mm'}, \tag{27}$$

where $\odot$ is the Hadamard element-wise product, and $s_{mm'}$ is zero except when $m = m' = k$ for $k = 1, \ldots, K$, in which case one has $s_{mm'} = -\eta\exp(-\eta\hat{a}_k - 1)$.

We then bound the spectrum of the Hessian matrix. Define

$$r_{mm'} = -\eta(D_m \odot D_{m'}) \cdot \frac{\hat{P}}{\sum_{ij}\hat{P}_{ij}} + \eta\frac{\left(\hat{P} \cdot D_m\right)\left(\hat{P} \cdot D_{m'}\right)}{\left(\sum_{ij}\hat{P}_{ij}\right)^2}, \tag{28}$$

and $R = [r_{mm'}]_{m,m'=1}^{K+L}$. Let $(I, J)$ be a random vector supported on $[n] \times [n]$, and moreover let $(I, J)$ follow the multinomial distribution with $\mathbb{P}[(I, J) = (i, j)] = \frac{\hat{P}_{ij}}{\sum_{ij}\hat{P}_{ij}}$. Moreover, for $m = 1, \ldots, K + L$, let $Y_m$ be the random variable defined by $Y_m = (D_m)_{IJ}$. Then, one can directly verify that $R = -\eta\mathrm{Cov}(Y_1, \ldots, Y_{K+L})$, and therefore $R$ is a negative semi-definite matrix. Let $S = [s_{mm'}]_{m,m'=1}^{K+L}$. The matrix $S$ is a diagonal matrix with non-positive entries and therefore

is likewise negative semi-definite. By the trace bound one has

$$\|R\|_2 \le - \sum_{m=1}^{K+L} r_{mm}$$

$$= \sum_{m=1}^{K+L} \eta \left( (D_m \odot D_m) \cdot \frac{\hat{P}}{\sum_{ij} \hat{P}_{ij}} - \left( \frac{\hat{P} \cdot D_m}{\sum_{ij} \hat{P}_{ij}} \right)^2 \right)$$

$$\le \sum_{m=1}^{K+L} \eta \left( (D_m \odot D_m) \cdot \frac{\hat{P}}{\sum_{ij} \hat{P}_{ij}} \right)$$

$$\le \eta (K + L) c_d^2,$$

and therefore one has the spectral bound $S - \eta(K+L)c_d^2 I \preceq \nabla_a^2 f_{\mathrm{aug}} \preceq S$.

For simplicity, for the current variable $(x, y, a)$, define a function $g$ by $g(\hat{a}) = f_{\mathrm{aug}}(x, y, \hat{a})$. By Taylor's remainder theorem, for any $\delta a$, one has

$$f_{\mathrm{aug}}(x, y, a + \delta a) - f_{\mathrm{aug}}(x, y, a) = g(a + \delta a) - g(a) = \delta a^\top \nabla_a g(a) + \frac{1}{2} \delta a^\top \nabla_a^2 g(a + \xi \delta a) \delta a,$$

where $\xi \in [0, 1]$ is an unknown quantity. Thus, the spectral bound on $\nabla_a^2 f_{\mathrm{aug}}$ leads to the following inequality,

$$f_{\mathrm{aug}}(x, y, a + \delta a) - f_{\mathrm{aug}}(x, y, a)$$

$$\ge \delta a^\top \nabla_a g(a) - \frac{1}{2} \eta(K+L) c_d^2 \|\delta a\|_2^2 - \frac{1}{2} \sum_{k=1}^{K} \eta \exp(-\eta a_k - 1) \exp(-\eta \xi \delta a_k) \delta a_k^2$$

$$= \sum_{m=1}^{K+L} \left( \delta a_m d_m - \frac{1}{2} \eta(K+L) c_d^2 \delta a_m^2 \right) - \frac{1}{2} \sum_{k=1}^{K} \eta \exp(-\eta a_k - 1) \exp(-\eta \xi \delta a_k) \delta a_k^2$$

where $d_m = \partial_{a_m} g(a)$ is due to $\sum_{ij} P_{ij} = 1$.

Importantly, the lower bound in equation 29 is fully separable in terms of the $\delta a_m$ terms. In what follows, we use equation 29 to give a construction for $\delta a$, which would then give a lower bound for the improvement in the Lyapunov function in the $a$ update step. For $l = 1, \ldots, L$, one sets $\delta a_{l+K} = \frac{d_{l+K}}{\eta(K+L)c_d^2}$. One then has

$$\delta a_{l+K} d_{l+K} - \frac{1}{2} \eta(K+L) c_d^2 \delta a_{l+K}^2 = \frac{d_{l+K}^2}{2\eta(K+L)c_d^2}.$$

For $k = 1, \ldots K$, one sets

$$\delta a_k = \max \left( -\frac{\log(2)}{\eta}, \frac{d_k}{\eta \left( 2 \exp(-\eta a_k - 1) + (K+L)c_d^2 \right)} \right). \tag{30}$$

We shall prove that the construction of $\delta a_k$ leads to the following bound:

$$\delta a_k d_k - \frac{1}{2} \eta(K+L) c_d^2 \delta a_k^2 - \frac{1}{2} \eta \exp(-\eta a_k - 1) \exp(-\eta \xi \delta a_k) \delta a_k^2 \ge |d_k| \min \left( \frac{1}{8\eta}, \frac{|d_k|}{8\eta c_d + 4\eta(K+L)c_d^2} \right).$$

By construction, one has $\exp(-\eta \xi \delta a_k) \le 2$, and thus

$$-\frac{1}{2} \eta \exp(-\eta a_k - 1) \exp(-\eta \xi \delta a_k) \delta a_k^2 \ge -\frac{1}{2} \left( 2\eta \exp(-\eta a_k - 1) \delta \right) a_k^2.$$

If $-\frac{\log(2)}{\eta} \ge \frac{d_k}{\eta \left( 2 \exp(-\eta a_k - 1) + (K+L)c_d^2 \right)}$, then one has $\delta a_k = -\frac{\log(2)}{\eta} \le 0$ and $d_k \le 0$. By equation 30 one has

$$\delta a_k \eta \left( 2 \exp(-\eta a_k - 1) + (K+L)c_d^2 \right) \ge d_k,$$

and thus multiplying both sides by $-\frac{1}{2}\delta a_k$, the inequality becomes

$$-\frac{1}{2}\eta\left(2\exp(-\eta a_k - 1) + (K+L)c_d^2\right)\delta a_k^2 \geq -\frac{1}{2}\delta a_k d_k.$$

Thus

$$\delta a_k d_k - \frac{1}{2}\eta(K+L)c_d^2\delta a_k^2 - \frac{1}{2}\eta\exp(-\eta a_k - 1)\exp(-\eta\xi\delta a_k)\delta a_k^2$$

$$\geq \delta a_k d_k - \frac{1}{2}\eta\left(2\exp(-\eta a_k - 1) + (K+L)c_d^2\right)\delta a_k^2$$

$$\geq \delta a_k d_k - \frac{1}{2}\delta a_k d_k$$

$$= \frac{1}{2}\delta a_k d_k$$

$$= |d_k|\frac{\log(2)}{2\eta},$$

which in particular implies the claimed bound as $\frac{\log(2)}{2} \geq \frac{1}{8}$.

Otherwise, if $-\frac{\log(2)}{\eta} \leq \frac{d_k}{\eta\left(2\exp(-\eta a_k - 1) + (K+L)c_d^2\right)}$, then one has $\delta a_k = \frac{d_k}{\eta\left(2\exp(-\eta a_k - 1) + (K+L)c_d^2\right)}$ by construction. Then, one has

$$\delta a_k d_k - \frac{1}{2}\eta(K+L)c_d^2\delta a_k^2 - \frac{1}{2}\eta\exp(-\eta a_k - 1)\exp(-\eta\xi\delta a_k)\delta a_k^2$$

$$\geq \delta a_k d_k - \frac{1}{2}\eta\left(2\exp(-\eta a_k - 1) + (K+L)c_d^2\right)\delta a_k^2$$

$$= \frac{d_k^2}{2\eta\left(2\exp(-\eta a_k - 1) + (K+L)c_d^2\right)}$$

$$= \frac{d_k^2}{4\eta\exp(-\eta a_k - 1) + 2\eta(K+L)c_d^2}$$

$$\geq |d_k|\frac{|d_k|}{4\eta|d_k| + 4\eta c_d + 2\eta(K+L)c_d^2}$$

$$\geq |d_k|\min\left(\frac{1}{8\eta}, \frac{|d_k|}{8\eta c_d + 4\eta(K+L)c_d^2}\right),$$

where the third inequality is obtained by applying the mediant inequality, and the second inequality is because

$$\exp(-\eta a_k - 1) \leq |\exp(-\eta a_k - 1) - P\cdot D_k| + |P\cdot D_k| = |d_k| + |P\cdot D_k| \leq |d_k| + c_d.$$

The proof for equation 25 is by organizing the arranged results:

$$f(x + t'\mathbf{1}, y, a') - f(x, y, a)$$

$$= f_{\text{aug}}(x, y, a') - f_{\text{aug}}(x, y, a)$$

$$= \max_{\tilde{a}} f_{\text{aug}}(x, y, \tilde{a}) - f_{\text{aug}}(x, y, a)$$

$$\geq f_{\text{aug}}(x, y, a + \delta a) - f_{\text{aug}}(x, y, a)$$

$$= g(a + \delta a) - g(a)$$

$$= \delta a^\top \nabla_a g(a) + \frac{1}{2}\delta a^\top \nabla_a^2 g(a + \xi\delta a)\delta a$$

$$\geq \sum_{k=1}^{K}\left(\delta a_k d_k - \frac{1}{2}\eta(K+L)c_d^2\delta a_k^2 - \frac{1}{2}\eta\exp(-\eta a_k - 1)\exp(-\eta\xi\delta a_k)\delta a_k^2\right)$$

$$+ \sum_{l=1}^{L}\left(\delta a_{l+K}d_{l+K} - \frac{1}{2}\eta(K+L)c_d^2\delta a_{l+K}^2\right)$$

$$\geq \sum_{k=1}^{K}|d_k|\min\left(\frac{1}{8\eta}, \frac{|d_k|}{8\eta c_d + 4\eta(K+L)c_d^2}\right) + \sum_{l=1}^{L}\frac{d_{l+K}^2}{2\eta(K+L)c_d^2}.$$

$\square$

*Remark* 1. In the proof for Theorem 3, it might be advantageous to use an alternative spectral bound when the matrices $D_m$ have special sparsity structures. In such special cases, it is better to use the Gershgorin circle theorem instead of the trace bound in the numerical treatment for the $a$ step.

One defines

$$r_{1;mm'} = -\eta(D_m \odot D_{m'}) \cdot \frac{\hat{P}}{\sum_{ij} \hat{P}_{ij}},$$

$$r_{2;mm'} = \eta \frac{\left(\hat{P} \cdot D_m\right)\left(\hat{P} \cdot D_{m'}\right)}{\left(\sum_{ij} \hat{P}_{ij}\right)^2}, \tag{31}$$

and let $R_1 = [r_{1;mm'}]_{m,m'=1}^{K+L}, R_2 = [r_{2;mm'}]_{m,m'=1}^{K+L}$. Let $Y_m$ for $m = 1, \ldots, K + L$ be as in the proof of Theorem 3. One can directly verify that $R_2$ is positive semi-definite as it is the outer product of the vector $\sqrt{\eta}\mathbb{E}[Y_1, \ldots, Y_{K+L}]$ with itself, and thus $S + R_1 \preceq \nabla_a^2 f_{\text{aug}}$. By the Gershgorin circle theorem, it follows that $\|R_1\|_2$ is bounded by the matrix 1-norm of $R_1$. In particular, define $E \in \mathbb{R}^{n \times n}$ as the sum of the constraint matrices up to absolute value, i.e.

$$(E)_{ij} = \sum_{m=1}^{K+L} |(D_m)_{ij}|.$$

Then, one has

$$\|R_1\|_2 \leq \max_m \sum_{m'=1}^{K+L} |r_{1;mm'}|$$

$$= \eta \max_m \sum_{m'=1}^{K+L} \left|(D_m \odot D_{m'}) \cdot \frac{\hat{P}}{\sum_{ij} \hat{P}_{ij}}\right|$$

$$= \eta \max_m \sum_{m'=1}^{K+L} \left|\left(\frac{\hat{P}}{\sum_{ij} \hat{P}_{ij}} \odot D_{m'}\right) \cdot D_m\right|$$

$$\leq \eta c_d \sum_{m'=1}^{K+L} \|\frac{\hat{P}}{\sum_{ij} \hat{P}_{ij}} \odot D_{m'}\|_1$$

$$= \eta c_d \sum_{m'=1}^{K+L} \left|\frac{\hat{P}}{\sum_{ij} \hat{P}_{ij}} \cdot D_{m'}\right|$$

$$\leq \eta c_d \left(\frac{\hat{P}}{\sum_{ij} \hat{P}_{ij}} \cdot E\right)$$

$$\leq \eta c_d \|E\|_\infty,$$

where the second and third equality is by the property of the Hadamard product, the second and fourth inequality is by Holder's inequality, and the third inequality is by the definition of $E$. We define $c_e = \|E\|_\infty$, which gives one the following bound in contrast to equation 29:

$$f_{\text{aug}}(x, y, a + \delta a) - f_{\text{aug}}(x, y, a)$$

$$\geq \delta a^\top \nabla_a g(a) - \frac{1}{2}\eta c_e c_d \|\delta a\|_2^2 - \frac{1}{2}\sum_{k=1}^{K} \eta \exp(-\eta a_k - 1)\exp(-\eta\xi\delta a_k)\delta a_k^2 \tag{32}$$

$$= \sum_{m=1}^{K+L}\left(\delta a_m d_m - \frac{1}{2}\eta c_e c_d \delta a_m^2\right) - \frac{1}{2}\sum_{k=1}^{K} \eta \exp(-\eta a_k - 1)\exp(-\eta\xi\delta a_k)\delta a_k^2.$$

Using equation 32, one can derive a tighter bound in the conclusion part of Theorem 3 by applying a similar argument in the main statement. Overall, the bound in equation 32 is more advantageous when $c_e$ is significantly smaller than $(K + L)c_d$, and indeed one can see that $c_e \leq (K + L)c_d$. One situation in which an advantage exists is when the constraints $D_m$ are themselves sparse, which is the case in capacity constrained OT.

We now prove Theorem 2.

*Proof.* For any iteration index $u$, let $Q_x^u, Q_y^u, Q_a^u$ be the quantity as defined by the Theorem for $P_u$ in place of $P$. Similarly we define $d_m^u$ by equation 24 with $P_u$ in place of $P$. One then has by construction that

$$f(x^{u+1}, y^{u+1}, a^{u+1}) - f(x^u, y^u, a^u) \geq \frac{1}{3} \left( Q_x^u + Q_y^u + Q_a^u \right). \tag{33}$$

In particular, for any $k = 1, \ldots, K$, one has

$$f(x^{u+1}, y^{u+1}, a^{u+1}) - f(x^u, y^u, a^u) \geq \frac{1}{3} |d_k^u| \min \left( \frac{1}{8\eta}, \frac{|d_k^u|}{8\eta c_d + 4\eta(K+L)c_d^2} \right).$$

Suppose that there are $N_I$ iterations for which $\frac{1}{8\eta} \leq \frac{|d_k^u|}{8\eta c_d + 4\eta(K+L)c_d^2}$ for some $k = 1, \ldots, K$. For such $u$, one has $|d_k^u| = \Omega(1)$ and therefore one has $f(x^{u+1}, y^{u+1}, a^{u+1}) - f(x^u, y^u, a^u) = \Omega(1)$. Thus $N_I = O(c_g)$, and therefore for $k = 1, \ldots, K$, one has $\frac{1}{8\eta} \geq \frac{|d_k^u|}{8\eta c_d + 4\eta(K+L)c_d^2}$ except for $O(c_g)$ iterations. Let $C = \frac{1}{8\eta c_d + 4\eta(K+L)c_d^2}$, and then the following condition holds except for $N_I$ iterations:

$$f(x^{u+1}, y^{u+1}, a^{u+1}) - f(x^u, y^u, a^u) \geq \frac{1}{3} \left( \mathrm{KL} \left( P_u \mathbf{1} || r \right) + \mathrm{KL} \left( P_u^\top \mathbf{1} || c \right) + C \sum_{m=1}^{K+L} (d_m^u)^2 \right). \tag{34}$$

Let $N_{II}$ be the number of iterations for which the following condition is satisfied

$$\|\nabla f\|_1 = \|P_u \mathbf{1} - r\|_1 + \|P_u^\top \mathbf{1} - c\|_1 + \sum_{m=1}^{K+L} |d_m| > \epsilon. \tag{35}$$

Then, if $u$ is among the said $N_{II}$ iterations, one has

$$\epsilon^2 < \left( \|P_u \mathbf{1} - r\|_1 + \|P_u^\top \mathbf{1} - c\|_1 + \sum_{m=1}^{K+L} |d_m^u| \right)^2$$

$$\leq (K+L+2)^2 \left( \|P_u \mathbf{1} - r\|_1^2 + \|P_u^\top \mathbf{1} - c\|_1^2 + \sum_{m=1}^{K+L} (d_m^u)^2 \right)$$

$$\leq (K+L+2)^2 \left( 2\mathrm{KL} \left( P_u \mathbf{1} || r \right) + 2\mathrm{KL} \left( P_u^\top \mathbf{1} || c \right) + \sum_{m=1}^{K+L} (d_m^u)^2 \right),$$

where the second inequality is by Cauchy-Schwartz, the third inequality is equation 24, and the third inequality is by Pinsker's inequality. Thus, for at least $N_{II} - N_I$ iterations, both equation 34 and equation 35 are satisfied, under which there exists a constant $C'$ for which one has

$$\epsilon^2 \leq (K+L+2)^2 \left( 2\mathrm{KL} \left( P_u \mathbf{1} || r \right) + 2\mathrm{KL} \left( P_u^\top \mathbf{1} || c \right) + \sum_{m=1}^{K+L} (d_m^u)^2 \right)$$

$$\leq C' \left( f(x^{u+1}, y^{u+1}, a^{u+1}) - f(x^u, y^u, a^u) \right)$$

As $\sum_{u=0}^{\infty} f(x^{u+1}, y^{u+1}, a^{u+1}) - f(x^u, y^u, a^u) \leq c_g$, one must have $N_{II} - N_I = O\left( (\epsilon)^{-2} c_g \right)$. One then has $N_{II} = N_I + O\left( (\epsilon)^{-2} c_g \right) = O\left( (\epsilon)^{-2} c_g \right)$, as desired, and thus the first $u$ for which equation 11 holds occurs after $O\left( (\epsilon)^{-2} c_g \right)$ iterations.

Lastly, equation 12 holds as a result of equation 11 because one has $|d_{K+l}^u| = P^u \cdot D_{K+l}$ and $|d_k^u| \geq |\min(0, P^u \cdot D_k)|$.

$\square$

As a result of Theorem 2, one has the following bound if one applies a rounding operation to the output of $P$ in Theorem 2:

**Proposition 1.** *Let $P$ be the same as in Theorem 2, let $\mathcal{U}_{r,c}$ denote the set of transport matrix from $r$ to $c$, and let $\mathrm{Round}(P,\mathcal{U}_{r,c})$ be the result of applying the rounding algorithm in Altschuler et al. (2017) to obtain projection of $P$ into $\mathcal{U}_{r,c}$. Define constraint violation of $P$ by*

$$\mathrm{Violation}(P) = \sum_{k=1}^{K} \left|\min\left(\mathrm{Round}(P,\mathcal{U}_{r,c}) \cdot D_k, 0\right)\right|$$

$$+ \sum_{l=1}^{L} \left|\mathrm{Round}(P,\mathcal{U}_{r,c}) \cdot D_{l+K}\right|.$$

*One has*

$$\mathrm{Violation}(P) \le \epsilon\left(1 + 2(K+L)c_d\right)$$

*where $c_d := \max_{m \in [K+L]} \|D_m\|_\infty$.*

*Proof.* This proof is a simple consequence of existing results. By Lemma 7 in Altschuler et al. (2017), one has

$$\|P - \mathrm{Round}(P,\mathcal{U}_{r,c})\|_1 \le 2\epsilon.$$

Thus one has

$$\mathrm{Violation}(P) \le \sum_{k=1}^{K} \left|\min\left(P \cdot D_k, 0\right)\right| + \sum_{l=1}^{L} \left|P \cdot D_{l+K}\right| + \sum_{m=1}^{K+L} \left|(P - \mathrm{Round}(P,\mathcal{U}_{r,c})) \cdot D_m\right|$$

$$\le \epsilon + 2(K+L)c_d\epsilon,$$

where the last inequality uses Holder's inequality and equation 12. $\square$

## H   EQUIVALENCE OF PRIMAL AND PRIMAL-DUAL FORM

We now show that the primal form in equation 5 can be obtained from the primal-dual form by eliminating the dual variables.

**Proposition 2.** *Define*

$$L(P, s, x, y, a) = \frac{1}{\eta} P \cdot \log P + C \cdot P - x \cdot (P\mathbf{1} - r) - y \cdot (P^\top \mathbf{1} - c)$$

$$+ \frac{1}{\eta} \sum_{k=1}^{K} s_k \log s_k + \sum_{k=1}^{K} a_k s_k - \sum_{m=1}^{K+L} a_m(D_m \cdot P),$$

*and then, for $\mathcal{S}$ as in equation 4, the following equation holds:*

$$\max_{x,y,a} \min_{P,s} L(P, s, x, y, a, b) = \min_{P:P\mathbf{1}=r, P^\top\mathbf{1}=c, P\in\mathcal{S}} \frac{1}{\eta} P \cdot \log P + \sum_{k=1}^{K} \frac{1}{\eta}(D_k \cdot P) \cdot \log(D_k \cdot P) + C \cdot P \tag{36}$$

*Moreover, for the Lyapunov potential function $f$ in equation 7, one has*

$$f(x, y, a) = \min_{P,s} L(P, s, x, y, a). \tag{37}$$

*Proof.* As $L$ is concave in $x, y$ and convex in $P$, one can invoke the minimax theorem to interchange the operations of maximization and minimization. Therefore:

$$\min_{P,s} \max_{x,y,a} L(P, s, x, y, a) = \min_{P,s:P\mathbf{1}=r, P^\top\mathbf{1}=c, P\in\mathcal{S}, s_k=D_k\cdot P \,\forall k\in[K]} \frac{1}{\eta} P \cdot \log P + \frac{1}{\eta} s \cdot \log s + C \cdot P$$

$$= \min_{P:P\mathbf{1}=r, P^\top\mathbf{1}=c, P\in\mathcal{S}} \frac{1}{\eta} P \cdot \log P + \sum_{k=1}^{K} \frac{1}{\eta}(D_k \cdot P) \cdot \log(D_k \cdot P) + C \cdot P.$$

In terms of entries, one writes $L(P, s, x, y, a)$ as follows:

$$\max_{x_i, y_j} \min_{p_{ij}} L(P, s, x, y, a) = \frac{1}{\eta} \sum_{ij} p_{ij} \log p_{ij} + \sum_{ij} C_{ij} p_{ij} - \sum_i x_i (\sum_j p_{ij} - r_i) - \sum_j y_j (\sum_i p_{ij} - c_j)$$

$$+ \frac{1}{\eta} \sum_{k=1}^{K} s_k \log s_k + \sum_{k=1}^{K} a_k s_k - \sum_{m=1}^{K+L} \sum_{ij} a_m (D_m)_{ij} p_{ij}$$

We then solve the inner min problem explicitly by taking the derivative of $p_{ij}, s_k$ to zero, from which one obtains

$$p_{ij} = \exp(\eta(-C_{ij} + \sum_{m=1}^{K+L} a_m (D_m)_{ij} + x_i + y_j) - 1).$$

and

$$s_k = \exp(-\eta a_k - 1).$$

Plugging in the formula for $p_{ij}$ and $s_k$, one has

$$\min_{P,s} L(P, s, x, y, a)$$

$$= -\frac{1}{\eta} \sum_{ij} \exp(\eta(-C_{ij} + \sum_{m=1}^{K+L} a_m (D_m)_{ij} + x_i + y_j) - 1) + \sum_i r_i x_i + \sum_j c_j y_j - \frac{1}{\eta} \sum_{k=1}^{K} \exp(-\eta a_k - 1),$$

which is equal to $f(x, y, a)$. $\qquad\square$

## I  RELATION OF ALGORITHM 1 TO BREGMAN PROJECTION

In this section, we show that the proposed $a$ update step is equivalent to a Bregman projection for equality constraints. The Bregman projection step introduced in Benamou et al. (2015) introduces an iterative projection-based approach. As in the setting of Section 3, let $(x, y, a)$ be the current dual variable and let $P$ be the intermediate matrix corresponding to $(x, y, a)$.

Suppose that $K = 0$ and thus there are no inequality constraints. As in main text, we define $\mathcal{E}$ by the following space

$$\mathcal{E} := \bigcap_{l=1,\dots,L} \{M \mid D_l \cdot M = 0\}.$$

Moreover, assume $\sum_i r_i = \sum_j c_j = 1$ and define $\Delta_{n \times n}$ as the $n^2$-dimensional simplex. Let $a', t' = \arg\max_{\tilde{a}, \tilde{t}} f(x + \tilde{t}\mathbf{1}, y, \tilde{a})$, and let $P'$ be the intermediate matrix corresponding to $(x + t'\mathbf{1}, y, a')$. Then, the claimed equivalence can be seen by proving the following equation:

$$P' = \arg\min_{M \in \Delta_{n \times n} \cap \mathcal{E}} \text{KL}\,(M||P) \tag{38}$$

where for two entry-wise non-negative matrices $M, N$, the term $\text{KL}\,(M||N)$ is defined by $\text{KL}\,(M||N) = \sum_{ij} m_{ij} \left(\log(\frac{m_{ij}}{n_{ij}}) - 1\right)$. Suppose $M = [m_{ij}]_{i,j=1}^n$ achieves the optimality condition set in equation 38. As it is a constrained optimization problem, the necessary condition for optimality is that there exists $\mu$ and $\lambda_l$ for $l = 1, \dots, L$, so that the following holds:

$$\forall i, j \in [n], \quad \partial_{m_{ij}} \text{KL}\,(M||P) = \eta \left(\sum_{l=1}^{L} \lambda_l (D_l)_{ij} + \mu\right).$$

Utilizing the definition of $\text{KL}\,(\cdot||\cdot)$, one rewrites the above equation as below:

$$\log(m_{ij}) - \log(p_{ij}) = \eta \left(\sum_{l=1}^{L} \lambda_l (D_l)_{ij} + \mu\right).$$

Thus, there exists $\mu$ and $\lambda_l$ for $l = 1, \dots, L$, for which one has $M = \exp(\sum_l \lambda_l D_m + \mu \mathbf{1}\mathbf{1}^\top) \odot P$, where $\odot$ is the Hadamard product. Thus, one has

$$M_{ij} = \exp\left(\eta(-C_{ij} + \sum_{l=1}^{L} (a_m + \lambda_l)(D_l)_{ij} + x_i + y_j + \mu) - 1\right). \tag{39}$$

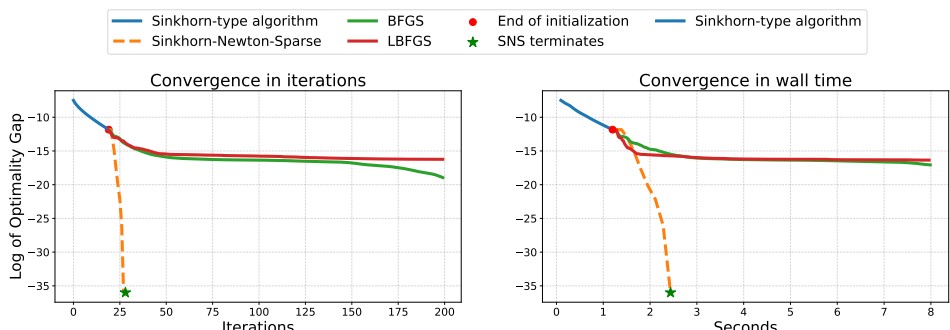

Figure 13: Performance of Quasi-Newton methods, compared against the Sinkhorn-Newton-Sparse algorithm and the Sinkhorn algorithm.

Furthermore, $M$ satisfies the following normalization condition

$$\sum_{ij} M_{ij} = 1. \tag{40}$$

Moreover, for $l \in [L]$, the following holds:

$$M \cdot D_l = 0. \tag{41}$$

As seen in the main text, $P'$ satisfies equation 40, equation 41. Moreover, by setting $\lambda_l = a'_m - a_m$ and $\mu = t'$, one can show that $P'$ also satisfies equation 39. As $f$ is concave and the equality constraints are affine, this shows that $P'$ satisfies the optimality condition in that of the right-hand side of equation 38.

## J  COMPARISON BETWEEN SINKHORN-NEWTON-SPARSE WITH QUASI-NEWTON METHODS

This section presents the result of quasi-Newton algorithms (Nocedal and Wright, 1999) applied to the variational form of constrained optimal transport problems. We show that, while being a reasonable proposal for solving entropic optimal transport with second-order information, traditional quasi-Newton algorithms work poorly in practice for constrained optimal transport when compared with sparse Newton iterations. In short, a quasi-Newton algorithm can be obtained from Sinkhorn-Newton-Sparse by replacing the Hessian approximation step in Algorithm 2. Specifically, instead of sparse approximation, a quasi-Newton method approximates the Hessian matrix through the history of gradient information. In particular, we consider the Broyden–Fletcher–Goldfarb–Shanno (BFGS) algorithm and the limited-memory Broyden–Fletcher–Goldfarb–Shanno (L-BFGS) algorithm, which are two of the most widely used quasi-Newton methods.

We repeat the experiment settings in Section 5, and the result is shown in Figure 13. To ensure a fair comparison, the quasi-Newton candidate algorithms are given the same Sinkhorn initialization as in the SNS algorithm. For both quasi-Newton methods, the provided search directions can be such that the line search procedure fails to terminate. To ensure fairness of comparison in terms of number of iterations, in such failure modes we restart the quasi-Newton algorithm by keeping the current dual variable while resetting the initial guess of the Hessian matrix. As the plot shows, quasi-Newton algorithms are inferior in numerical performance when compared against the SNS algorithm.

