# OpenReview forum: "A Sinkhorn-type Algorithm for Constrained Optimal Transport"
_ICLR.cc/2025/Conference — Submitted to ICLR 2025_

### Official Review · Reviewer_iUzs · 2024-11-02

**Soundness:** 2
**Presentation:** 2
**Contribution:** 1
**Rating:** 3
**Confidence:** 4

**Summary:**

This paper studies the algorithm for solving the optimal transport problem under equality and inequality constraints. The authors derive a Sinkhorn-type algorithm to solve the constrained optimal transport problem.

**Strengths:**

The paper extends the Sinkhorn algorithm to unconstrained optimal transport problem to constrained optimal transport problem successfully.

**Weaknesses:**

1. The paper is hard to follow. The paper lists technical details and equations without further explanations.  For example, which is function L in Line 202? There are no further comments or interpretations of Theorem 2. Sometimes, the author refers to the equation in the appendix, for example, Line 297. The message is not direct. For example, what is Section 3.1 for? Merge some points in Section 1.1 to make the contributions clearer. It is highly recommended that the authors reorganize and polish the paper to make it accessible...

2. What is the motivation for focusing on the constrained optimal transport? Although it could be theoretically interesting to extend Sinkhorn to this kind of optimization problem, the author should make the significance of studying the problem clear.

3. What is the resulting computational complexity of the proposed Sinkhorn-type algorithm?  It seems that only polynomial time can be proved in the paper, which is not attractive enough. The complexity of the Sinkhorn-type algorithm for solving unconstrained optimal transport problems is O(n^2/\epsilon^2); it seems that a comparable complexity should be derived for constrained optimal transport problems.

**Questions:**

1. Line 187, what does depending only on the LP mean? It is not clear to me that a constant can depend on the linear optimization problem.

2. Line 244, parameter setting can simply your analysis instead of ensuring the efficiency of the algorithm. Can the author improve their justifications of K=O(1), K+L=O(1)?

3. Line 227, in terms of computational complexity, APDAGD can not outperform.

---

> ### Author Response · Authors · 2024-11-14
> **Reply to the reviewer (Part 1)**
>
> The authors thank the reviewer for the reviewer's time. We would like to address the technical and theoretical concerns raised by the reviewer. We anticipate changes to the manuscript, and so all equation number references and line references will be referring to the pre-discussion version submitted to this conference. If the response provides some clarity and is satisfactory to the reviewer, we kindly suggest that the reviewer consider an increase of the score.
>
> ### Comment 1
> *The paper is hard to follow. The paper lists technical details and equations without further explanations. For example, which is function L in Line 202? There are no further comments or interpretations of Theorem 2. Sometimes, the author refers to the equation in the appendix, for example, Line 297. The message is not direct. For example, what is Section 3.1 for? Merge some points in Section 1.1 to make the contributions clearer. It is highly recommended that the authors reorganize and polish the paper to make it accessible...*
>
> **Response:**
>
> We would love to have further feedbacks on the parts of the manuscript for which the reviewer thinks deserve enhanced clarification. We give some clarifying explanations here for the specific points raised.
>
> For the function $L$, the function is the primal-dual form, which combines the original function with additional terms taking account of the equality and inequality constraints. The term $L$ is sometimes referred to as the Lagrangian. Following the reviewer's suggestion, we have edited the text by further specifying that $L$ is the primal-dual form.
>
> For the Hessian matrix in line 297, the Hessian blocks would take quite a lot of space, and the main message is to show that the cost for running Algorithm 1 enjoys an \(O(n^2)\) scaling. We omit the formula for computing the Hessian so as to not overload the notation. Following the reviewer's suggestion, we substantially changed the wording for the paragraph to improve clarity. We thank the reviewer for this valuable suggestion.
>
> For the contribution section which is Section 1.1, we follow the reviewer's suggestion and moved some points from Section 3.1 to Section 1.1 to further motivate expert readers. To further address the reviewer's question, Section 3.1 involves intuitive though technically intricate acceleration techniques, and it is placed after Algorithm 1 in Section 3 as a clean exposition would not be possible before the technical build-up in Algorithm 1. We appreciate the reviewer's feedback and we thank the reviewer for this suggestion on Section 3.1 that improves the clarity of this manuscript.
> If the reviewer has more suggestions for reorganizing the paper, we would greatly appreciate any further suggestions.
>
> ### Comment 2
>
> *What is the motivation for focusing on the constrained optimal transport? Although it could be theoretically interesting to extend Sinkhorn to this kind of optimization problem, the author should make the significance of studying the problem clear.*
>
> **Response:**
> We would love to provide further contexts. As of right now, the field of entropic optimal transport largely focuses on the setting of unconstrained optimal transport, and it has sprung up a lot of applications, such as domain adaptation [1], model fusion [2], unsupervised learning [3], and computer vision [4].
>
> The motivation for constrained optimal transport typically is based on the fact that the cost matrix involved in such OT models may not be the best option. For example, in domain adaptation, the OT map concerns a transport between source data $\{x\}$ and target data $\{y\}$ and the overall goal is to have some alignment between the source and the target data through optimal transport. To model the cost matrix $C$, two natural choices for the distance function would be $c_1(x, y) = \lvert x - y \rvert_{2} $ and $c_2(x, y) = | x - y |_{1}$. One way to consider both distance functions is to linearly combine $c_1$ and $c_2$ to form a new distance function, and the constrained OT perspective would say that the OT coupling would need to optimize over $c_1$ while having some performance lower-bound for $c_2$, or vice-versa. Similarly, all of the aforementioned cases in [1-4] uses some form of distance function but a priori a canonical choice for which one is the best cost.
>
> Therefore, using constrained OT might lead to better coupling matrix design, which could further help with OT applications in machine learning. Importantly, the cost for Algorithm 1 is $O(n^2)$, which means that it is feasible to adapt the proposed Sinkhorn-type algorithm to practical machine learning settings, and it enjoys a good scaling that makes it compatible with large data sets. The scope of this manuscript is that this work is a paper for fast algorithms, and our goal is to put forward an algorithm which enables machine learning practitioners to have more flexibility in using OT.

---

> > ### Author Response · Authors · 2024-11-14
> > **Reply to the reviewer (Part 2)**
> >
> > ### Comment 3
> > *
> > What is the resulting computational complexity of the proposed Sinkhorn-type algorithm? It seems that only polynomial time can be proved in the paper, which is not attractive enough. The complexity of the Sinkhorn-type algorithm for solving unconstrained optimal transport problems is $O(n^2/\epsilon^2)$; it seems that a comparable complexity should be derived for constrained optimal transport problems.*
> >
> > **Response:**
> >
> >
> > The per-iteration cost for the algorithm is $O(n^2)$. We do not know of a scaling law in a desired accuracy $\varepsilon$, and the conclusion section does mention more work is to be done for the analysis of Algorithm 1. While a stronger conclusion is more desirable, the analysis in Altschuler et al in [6] cannot fully lead to the generalization to a convergence bound in the constrained case.
> >
> > Part of the goal of this work is to provide a general-purpose iterative Bregman projection algorithm which was at the time not fully explored by Benamou et al in [5] OT community. The numerical experiments we conduct in this work focus on a setting in which the entropic regularization term $\eta$ is large, and we numerically show that the iteration complexity for Algorithm 1 to converge is mild and comparable to the practical numerical performance of the Sinkhorn algorithm. Therefore, we agree with the reviewer and we do anticipate a more refined analysis on the constraint update step can show a comparable convergence bound. The authors hope that a wider use of the proposed method can attract more researchers into the analysis of algorithmic convergence for constrained OT under entropic regularization.

---

> > > ### Author Response · Authors · 2024-11-14
> > > **Reply to the reviewer (Part 3)**
> > >
> > > ### Question 1
> > >
> > > *Line 187, what does depending only on the LP mean? It is not clear to me that a constant can depend on the linear optimization problem.*
> > >
> > >
> > > **Response:**
> > >
> > > The term \(\Delta\) is the optimality gap between vertices of the feasible solution polytope. Thus this term does not have dependence on the entropic parameter $\eta$, and instead only depend on the specification of the linear programming problem.
> > >
> > >
> > >
> > > ### Question 2
> > >
> > > *Line 244, parameter setting can simply your analysis instead of ensuring the efficiency of the algorithm. Can the author improve their justifications of $K=O(1), K+L=O(1)$?*
> > >
> > > **Response:**
> > >
> > > The assumption $K + L = O(1)$ is critical to the \(O(n^2)\) guarantee. For large constraints, the constrained OT problem would be too far away from OT to admit nice problem structure that one could exploit. To obtain the hessian matrix for the $a$ variable, one would need a computational complexity of $O((K+L)^2n^2)$. Thus, the assumption for $ K+ L = O(1)$ is generally a necessary condition to ensure constrained OT has the same $O(n^2)$ iteration complexity as Sinkhorn in the unconstrained case.
> > >
> > >
> > > ### Question 3
> > > *Line 227, in terms of computational complexity, APDAGD can not outperform.*
> > >
> > > **Response:**
> > >
> > > The authors agree with the reviewer that Sinkhorn has a more attractive convergence guarantee in the later stage. The paper pioneering the APDAGD algorithm approach [7] claims a better performance over Sinkhorn with dimension-free convergence bound. The claim of APDAGD outperforming Sinkhorn overall has been proven false by Lin et al in [8] after studying some unexplained constants. However, the numerical performance in [7] does show anecdotal evidence of better performance at the initial stage. Thus APDAGD is chosen to be a benchmark to justify the Sinkhorn-type algorithm (Algorithm 1). Similarly to the OT case, Figure 3 in this manuscript shows that APDAGD does have a slight advantage over Algorithm 1 for the first few iterations, but the Sinkhorn algorithm outperforms after those initial stages. Following the reviewer's suggestion, we added clarifying remarks to the numerical experiment section for a comparison between APDAGD and the Sinkhorn-type algorithm.
> > >
> > >
> > > *We hope the response clears some of the reviewer's concern about the soundness of our work. If the response provides some clarity and is satisfactory to the reviewer, we kindly suggest that the reviewer consider an increase of the score.*
> > >
> > >
> > >
> > > ### References
> > >
> > > [1] Courty, N., Flamary, R., Habrard, A., & Rakotomamonjy, A. (2017). Joint distribution optimal transportation for domain adaptation. Advances in neural information processing systems, 30.
> > >
> > > [2] Singh, S. P., & Jaggi, M. (2020). Model fusion via optimal transport. Advances in Neural Information Processing Systems, 33, 22045-22055.
> > >
> > > [3] Cuturi, M. (2013). Sinkhorn distances: Lightspeed computation of optimal transport. Advances in neural information processing systems, 26.
> > >
> > > [4] Ge, Z., Liu, S., Li, Z., Yoshie, O., & Sun, J. (2021). Ota: Optimal transport assignment for object detection. In Proceedings of the IEEE/CVF conference on computer vision and pattern recognition (pp. 303-312).
> > >
> > > [5] Benamou, J. D., Carlier, G., Cuturi, M., Nenna, L., & Peyré, G. (2015). Iterative Bregman projections for regularized transportation problems. SIAM Journal on Scientific Computing, 37(2), A1111-A1138.
> > >
> > > [6] Altschuler, J., Niles-Weed, J., & Rigollet, P. (2017). Near-linear time approximation algorithms for optimal transport via Sinkhorn iteration. Advances in neural information processing systems, 30.
> > >
> > > [7] Dvurechensky, P., Gasnikov, A., & Kroshnin, A. (2018, July). Computational optimal transport: Complexity by accelerated gradient descent is better than by Sinkhorn’s algorithm. In International conference on machine learning (pp. 1367-1376). PMLR.
> > >
> > > [8] Lin, T., Ho, N., & Jordan, M. I. (2022). On the efficiency of entropic regularized algorithms for optimal transport. Journal of Machine Learning Research, 23(137), 1-42.

---

> > > > ### Comment · Reviewer_iUzs · 2024-11-23
> > > > **Response to the author**
> > > >
> > > > I appreciate the author’s detailed response. However, the motivation for studying constrained OT still remains unconvincing to me. Without a more advanced theoretical analysis of the scaling law in a desired accuracy, the contributions of this paper seem to be limited. Therefore, I will maintain my current score.

---

> > > > > ### Author Response · Authors · 2024-11-24
> > > > > **Reply to the Reviewer iUzs**
> > > > >
> > > > > We thank the reviewer for their feedback and for taking the time to assess the paper. We aim to address the points raised and kindly request further clarification to ensure that all concerns are fully addressed.
> > > > >
> > > > > **Soundness Rating** The current rating indicates "Soundness: 2 (Fair)." We believe we have thoroughly addressed all the technical questions previously mentioned by the reviewer. We kindly request the reviewer to specify any additional technical concerns or gaps in the analysis that we may have overlooked.
> > > > >
> > > > > **Presentation Rating** The current rating is "Presentation: 1 (Poor)." We would appreciate it if the reviewer could identify specific sections or aspects of the paper that were difficult to follow. Furthermore, if our earlier responses have clarified some of these issues, we kindly request confirmation from the reviewer.
> > > > >
> > > > > **Theoretical Analysis of Scaling Laws** The reviewer states: "Without a more advanced theoretical analysis of the scaling law in a desired accuracy, the contributions of this paper seem to be limited." We request clarification on the specific type of advanced analysis the reviewer is seeking. While we acknowledge the reviewer’s perspective, we are concerned that such a critique could be applied broadly to many works unless it is grounded in the specific context of the contributions and methodology presented in the paper.
> > > > >
> > > > > **Motivation for Studying Constrained OT** The reviewer comments: "However, the motivation for studying constrained OT still remains unconvincing to me." We recognize and respect this viewpoint but wish to highlight the significance of constrained optimal transport (OT) as established in the seminal work of [Benamou, Carlier, Cuturi, Nenna, Peyré], a highly regarded contribution to the computational OT literature. Our work builds directly on this foundational paper and introduces a general and robust algorithm for constrained OT, which can be readily applied to machine learning tasks such as the ranking case and the geometric transport case described in the paper.
> > > > >
> > > > > We hope the reviewer might **reconsider** the scoring. While it is reasonable to have doubts about the entire formulation of entropic OT in the constrained case, we believe that the reviewer’s concerns about the contributions of our work could equally be applied to [Benamou, Carlier, Cuturi, Nenna, Peyré]. In terms of contributions to the OT community, this work significantly strengthens a well-regarded approach, and we wonder if the reviewer’s concerns might be mitigated by this consideration.
> > > > >
> > > > > We also wish to summarize our contributions from a broader perspective. Assuming the reviewer agrees that constrained OT is scientifically significant, the next immediate observation is that relying on a linear programming package for this problem would be too slow and not scalable. The work of [Benamou, Carlier, Cuturi, Nenna, Peyré] provides a scalable formulation for constrained OT.
> > > > > Our work further shows that this approach can be generalizable and can have a practical runtime complexity that is a constant difference from Sinkhorn, and in fact our algorithm can deal with quite respectable problem sizes such as $n = 5000\sim50000$.

---

> > > > > > ### Author Response · Authors · 2024-11-26
> > > > > >
> > > > > > Dear Reviewer iUzs, we kindly request you to review the points we have raised above. As the deadline for making changes to the PDF is approaching, your timely feedback would be greatly appreciated.

---

### Official Review · Reviewer_kbiN · 2024-11-04

**Soundness:** 4
**Presentation:** 3
**Contribution:** 4
**Rating:** 8
**Confidence:** 4

**Summary:**

The paper presents Entropy regularization formulation and a Sinkhorn algorithm to solve a more generalized class of Optimal Transport (OT) problems --- specifically, OT problems with inequality and equality constraints.

The writing and presentation are clear and unambiguous for the most part. There is a significant theoretical contribution since the paper considers a larger class of problems than the ones typically considered in OT-related works. However, the motivation of the problem is lacking. Acceleration techniques that are presented in Section 3.1 are not necessarily original since they are an obvious extension of existing techniques. I think there is enough reason to accept the paper since it addresses an OT formulation that is rarely explored. More specifically, this solution is the first of its kind, and it will interest the ICLR audience. Having said that, I do have some reservations about giving a strong acceptance recommendation.

One of the listed contributions (on Partial Optimal Transport) is only mentioned briefly and not included in the main paper. This seems like an attempt to bypass the page limit constraints, which, in my opinion, should not be encouraged. However, I will leave that to the discretion of the editors.

**Strengths:**

- The paper takes initial steps in the natural direction of OT problem research.
- The theoretical results (such as the convergence rates and bounds) presented herein will likely be referenced in the foreseeable future. So, the results themselves are significant.

**Weaknesses:**

Firstly, the problem itself is not motivated well in the paper. The authors should consider working on the introduction to establish the relevance of this work better.

Secondly, the only novelty I see is in Algorithm 1. However, I consider the theoretical contribution itself significant enough to overlook this shortcoming.

**Questions:**

- What is $n$ in Theorem 1?
- It is not clear why $f$ is a Lyapunov function. Can the authors explain this part in more detail?

---

> ### Author Response · Authors · 2024-11-14
> **Reply to the reviewer (Part 1)**
>
> The authors thank the reviewer for the reviewer's time. We anticipate changes to the manuscript, and so all equation number references and line references will be referring to the pre-discussion version submitted to this conference.
> ### Comment 1
> *Firstly, the problem itself is not motivated well in the paper. The authors should consider working on the introduction to establish the relevance of this work better.*
>
> **Response:**
>
> The authors agree with the reviewer the text lacks motivation that could otherwise make the paper stronger. One limitation faced by this work is that as of right now, the field of entropic optimal transport largely focuses on the setting of unconstrained optimal transport. To the best of our knowledge, the constrained optimal transport is not currently widely considered in the setting of machine learning. As the paper primarily works on fast algorithms, the aim of this work is to provide a satisfactory answer for numerical treatment for constrained optimal transport, and we hope that this work enables machine learning practitioners to have more flexibility in using OT. The authors think a stronger motivation than what was written would not be sufficiently backed by existing research. One notable example of constrained OT used in machine learning is partial optimal transport (POT), and we show that our Algorithm extends nicely into POT and has very good performance.
>
> We propose a possible direction for constrained optimal transport in machine learning, and this is also hinted in the main text in the related work section.
> A typical use case for constrained optimal transport is when the cost matrix involved in the OT models has more than one good options. For example, in domain adaptation [1], the OT map concerns a transport between source data $\{x\}$ and target data $\{y\}$ and the overall goal is to have some alignment between the source and the target data through optimal transport. To model the cost matrix $C$, two natural choices for the distance function would be $c_1(x, y) = \lvert x - y \rvert_{2}$ and $c_2(x, y) = \lvert x - y \rvert_{1}$. One way to consider both distance functions is to linearly combine $c_1$ and $c_2$ to form a new distance function, and the constrained OT perspective would say that the OT coupling would need to optimize over $c_1$ while having some performance lower-bound for $c_2$, or vice-versa. This type of constrained optimal transport might help with machine learning practitioners obtain better coupling matrices for their down-stream applications. As the scaling of the Algorithm 1 is $O(n^2)$, there is indeed a lot of potential in using this Algorithm for machine learning.
>
>
> ### Comment 2
> *Secondly, the only novelty I see is in Algorithm 1. However, I consider the theoretical contribution itself significant enough to overlook this shortcoming.*
>
> **Response:**
>
> The reviewer thanks the reviewer for the comment. Part of the goal of this work is to provide a general-purpose iterative Bregman projection algorithm which was at the time not fully explored by Benamou et al [2] in the OT community. As the Benamou et al paper has a substantial time gap between now, a large part of the work is to integrate noteworthy ideas from optimal transport to the constrained setting. Incorporating the existing ideas are in itself a rather significant undertaking, and the author agree that the main novelty of this work lies in Algorithm 1 and in the treatment of inequality constraints by an entropic barrier.
>
> We list out two noteworthy novelties of the work. First, while the authors also agree that the acceleration techniques are straightforward extensions of existing work, part of the novelty is in figuring out which techniques are more suitable. One important goal of this work is to consider and compare all noteworthy practical OT acceleration techniques that can be extended to the general setting of unconstrained OT, and we show through numerical experiments that the most reliable techniques for successful implementations are entropic regularization scheduling and sparse Newton iterations. Other techniques such as APDAGD or quasi-Newton methods (BFGS and L-BFGS) are shown to be less suitable.
>
> Moreover, one noteworthy contribution of this work is our numerical treatment for partial optimal transport, which is excluded from the main text as it is a different formulation from the constrained OT considered in the main text. We show that likewise the Siknhorn-type algorithm and the sparse Newton iteration provides great advantage to the APDAGD algorithm considered in [3].

---

> > ### Author Response · Authors · 2024-11-14
> > **Reply to the reviewer (Part 2)**
> >
> > ### Question 1
> >
> > *What is $n$ in Theorem 1?*
> >
> > **Response:**
> >
> > The $n$ term in this work is used exclusively as the system size. Following the reviwer's suggestion, we changed the statement in Theorem 1 to remind the reader that the row and column constraints are respectively indexed by $i \in [n] =\{1, \ldots, n\}$ and $j \in [n]$, and we added a explanation of $n$ in the beginning of Section 2 (the "notation" paragraph) for better clarification. The authors thank the reviewer for the clarification question.
> >
> >
> >
> > ### Question 2
> >
> > *It is not clear why $f$ is a Lyapunov function. Can the authors explain this part in more detail?*
> >
> > **Response:**
> >
> > The term Lyapunov function is used here to refer to the dual potential. We show that $f$ is obtained by performing minimization over the primal variable $P, s$ over the Lagrangian term $L$, and $f$ is therefore the dual potential. For the setting of Sinkhorn, the dual potential function $f$ is referred to as the Lyapunov potential function (such as in Altschuler et al [4]), and we use the term here for consistency with the typical word choice in this field.
> >
> >
> > ### References
> >
> > [1] Courty, N., Flamary, R., Habrard, A., & Rakotomamonjy, A. (2017). Joint distribution optimal transportation for domain adaptation. Advances in neural information processing systems, 30.
> >
> > [2] Benamou, J. D., Carlier, G., Cuturi, M., Nenna, L., & Peyré, G. (2015). Iterative Bregman projections for regularized transportation problems. SIAM Journal on Scientific Computing, 37(2), A1111-A1138.
> >
> > [3] Nguyen, A. D., Nguyen, T. D., Nguyen, Q. M., Nguyen, H. H., Nguyen, L. M., & Toh, K. C. (2024, March). On partial optimal transport: Revising the infeasibility of sinkhorn and efficient gradient methods. In Proceedings of the AAAI Conference on Artificial Intelligence (Vol. 38, No. 8, pp. 8090-8098).
> >
> > [4] Altschuler, J., Niles-Weed, J., & Rigollet, P. (2017). Near-linear time approximation algorithms for optimal transport via Sinkhorn iteration. Advances in neural information processing systems, 30.

---

> > > ### Comment · Reviewer_kbiN · 2024-11-24
> > >
> > > I believe the authors have adequately responded to my queries. However, I will keep the score as it is because the contributions/implications of this work themselves are not significant enough to warrant a higher score.

---

### Official Review · Reviewer_C23h · 2024-11-04

**Soundness:** 4
**Presentation:** 3
**Contribution:** 2
**Rating:** 6
**Confidence:** 3

**Summary:**

This paper addresses scalable solutions to the problem of optimal transport by Kantorovich relaxation, under arbitrary additional linear constraints. By introducing a set of slack variables and adopting an entropic regularization, it arrives at a dual formulation, which is then solved by a block coordinate descent algorithm with three blocks. Two of them can be exactly optimized leading to a Sinkhorn-type procedure, while the third block, corresponding to the additional inequality constraints is proposed to be solved by the Newton's method. Some discussions  about accelerating the Newton's method is provided on account of the sparsity of the Hessian.

On the theoretical side, the paper shows two main results: first the fact that entropic regularization has an exponentially decreasing effect on the solution, w.r.t. the regularization parameter and second a guarantee on a sublinear rate of convergence of the algorithm.

The results are finally verified in some numerical experiments.

**Strengths:**

Constrained optimal transport may have various applications. IN ML it can be used e.g. for various domain adaptation scenarios with structured data. However, OT is a difficult problem to solve at scale. The Sinkhorn-type algorithms are widely believed to be a suitable way to address the difficulties with the exact OT problem. The paper pursues such a solution and the discussion is supported by rigorous mathematical results.  The numerical results also reflect the superior convergence properties of the proposed algorithm.

**Weaknesses:**

I generally find the paper interesting, but have few concerns regarding the motivation of the problem, the justifications of the algorithmic choices and the implications of the theoretical results. These are mentioned as questions in the next part.

**Questions:**

I generally find the paper interesting, but have few concerns:

1- My main concern is that I am not sure if the choice of an entropy regularization makes sense for the additional constraints. For the standard OT problem, this choice is justified as the dual problem can be solved by exact block coordinate descent, but if one needs to employ the Newton's method, then, why should not one use e.g. a logarithmic barrier (instead of entropic regularization), which has the self-concordance property and is potentially better for the doubling strategy.

2- Most of the analysis seems like a straightforward extensions of previous works Weed (2018) and Altschuler et al. (2017). A general concern about Theorem 1 is that although it is formulated as an exponential decay, it really shows the requirement that the regularization parameter grows proportionally with the inverse of the duality gap. In practice, the gap can be extremely small for large problems, leading to extremely large regularization parameters. On the other hand, the result of Theorem 2 seems to be dimension-free, but it actually depends on the dimension through the regularization parameter. As such, I think that Theorem 2 should clearly reflect the dependency on eta (which is hopefully linear as in Altschuler et al. (2017)).

3- Although the authors mention some applications of their problem in their literature review, I think that the paper generally does not well motivate the study. The experiments are on toy scenarios and do not reflect scalability as they consider relatively small problems. For the MNIST case, for example, a more realistic domain adaptation scenario could be considered with at least few thousand points in each domain.

---

> ### Author Response · Authors · 2024-11-14
> **Reply to the reviewer (Part 1)**
>
> The authors thank the reviewer for the reviewer's time. We would like to address the technical and theoretical concerns raised by the reviewer. We anticipate changes to the manuscript, and so all equation number references and line references will be referring to the pre-discussion version submitted to this conference. If the response provides some clarity and is satisfactory to the reviewer, we kindly suggest that the reviewer consider an increase of the score.
>
>
> ### Comment 1
> *My main concern is that I am not sure if the choice of an entropy regularization makes sense for the additional constraints. For the standard OT problem, this choice is justified as the dual problem can be solved by exact block coordinate descent, but if one needs to employ the Newton's method, then, why should not one use e.g. a logarithmic barrier (instead of entropic regularization), which has the self-concordance property and is potentially better for the doubling strategy.*
>
> **Response:**
> A rather significant benefit of entropic regularization is that this formulation allows for convergence guarantee by applying the analysis Weed (2018). The authors considered and experimented with a logarithmic barrier, and the performance for inequality constraint is comparable. Additionally, the authors were not able to find a good theoretical framework to analyze an entropic barrier for the coupling matrix while having a log barrier for the inequality constraint. The main work of this text still considers a simplified fixed parameter setting. In contrast, the log barrier framework would necessitate the doubling strategy, as it is typically not possible to achieve exponential convergence in the log barrier case without exponentially tuning the regularization strength. Therefore, the manuscript chooses the entropic barrier as it is much simpler for practitioners for implementation, while having no apparent numerical downside that the authors know of.
>
> Overall, the authors agree with the reviewer that the log barrier is a fruitful direction for further investigation, and the authors' position on barrier function was also highlighted in the Conclusion section of this manuscript.
>
>
>
>
> ### Comment 2
> *Most of the analysis seems like a straightforward extensions of previous works Weed (2018) and Altschuler et al. (2017). A general concern about Theorem 1 is that although it is formulated as an exponential decay, it really shows the requirement that the regularization parameter grows proportionally with the inverse of the duality gap. In practice, the gap can be extremely small for large problems, leading to extremely large regularization parameters. On the other hand, the result of Theorem 2 seems to be dimension-free, but it actually depends on the dimension through the regularization parameter. As such, I think that Theorem 2 should clearly reflect the dependency on eta (which is hopefully linear as in Altschuler et al. (2017)).*
>
>
> **Response:**
>
> The authors agree with the reviewer and wish to comment that the dependence on the optimality gap is an unavoidable consequence of Weed (2018). The entropic LP setting has a natural "slow rate" of $O(1/\eta)$ convergence to the true LP cost, which would be robust to the optimality gap.
>
> As for the algorithmic convergence in Theorem 2, we do not have a concrete answer for the dependence on $\eta$, as the term $c_g$ also has dependence on $\eta$, and we are currently unsure about the scaling, as the variation in the $a$ variable doesn't have a nice mathematical behavior as would be true $x, y$. However, without the dependence on $c_g$, we would have an $O(c_g \varepsilon^{-2}\eta)$ iteration complexity. We hypothesize that same as the case in Altschuler et al (2017), we would have $c_g = O(\eta)$ under reasonable assumptions, but
>
> We can show this scaling by modifying line 1552-1555 of the main text by taking out the constant's dependence on $\eta$. In the proof, the constant $C'$ has a dependence on $\eta$. Explicitly writing down the $\eta$ dependence would show
> $$
> \begin{aligned}
>     \epsilon^2
>     \leq &(K+L+2)^2\left(2 \mathrm{KL}(P_u\mathbf{1} || r) + 2\mathrm{KL}(P_u^{\top}\mathbf{1} || c) + \sum_{m=1}^{K+L}(d_m^u)^2\right)\\
>     \leq & O(\eta)\left(f(x^{u+1}, y^{u+1}, a^{u+1}) - f(x^{u}, y^{u}, a^{u}) \right),
> \end{aligned}
> $$
> and one could follow the rest of the proof and show the iteration complexity is $O(c_g \varepsilon^{-2}\eta)$. Assuming $c_g = O(\eta)$, this would be a quadratic dependence on $\eta$, and the numerical performance doesn't show any bad numerical behavior when $\eta$ is large. In fact, the regularization parameter $\eta$ in this work is typically chosen to be large to showcase performance. The analysis in this work does not focus on the $\eta$ dependence, and the authors suspect that a different technique focusing on the $\eta$ dependence would show a linear convergence bound that we observe from the empirical observation.

---

> > ### Author Response · Authors · 2024-11-14
> > **Reply to the reviewer (Part 2)**
> >
> > ### Comment 3
> >
> > *Although the authors mention some applications of their problem in their literature review, I think that the paper generally does not well motivate the study. The experiments are on toy scenarios and do not reflect scalability as they consider relatively small problems. For the MNIST case, for example, a more realistic domain adaptation scenario could be considered with at least few thousand points in each domain.*
> >
> > **Response:**
> >
> > Algorithm 1 and Algorithm 2 do have a $O(n^2)$ scaling. For comparable problems, when one has $K = 1$ and $L = 1$ (one equality and one inequality constraint) the run-time complexity of Algorithm 1 would take 2-3 times the compute time as that of the Sinkhorn iteration, and so the performance of the Algorithm 1 is quite robust. The MNIST dataset itself consists of images of 28*28 pixel, and so we were not able to use a bigger size image without making design choices.
> >
> > To address the reviewer's concern, the authors will conduct further experiments of larger random assignment problems with constraints. For unconstrained OT, the random assignment problem is known to be harder instances of optimal transport. We will provide an update to the paper before the end of the discussion period.

---

> > > ### Author Response · Authors · 2024-11-23
> > > **Update on numerical experiment**
> > >
> > > We are delighted to inform the reviewer that we have updated the manuscript on a larger random assignment instance. For the problem size of $n = 5000$, the proposed Sinkhorn-type and Sinkhorn-Newton-Sparse algorithms were able to converge to the entropically optimal solution in machine accuracy. The experiment shows that the proposed algorithm can scale to very large assignment problems.
> > >
> > > ## Result summary
> > > The Sinkhorn-type algorithm takes 40 iterations and ~80 seconds to converge. The Sinkhorn-Newton-Sparse (SNS) algorithm takes 21 iterations and ~45 seconds to converge. The algorithms are able to achieve good accuracy relatively quickly. In contrast, the case of $n=500$ takes 4 seconds for the SNS algorithm to converge. Thus, we show that the numerical performance even outperforms over the $O(n^2)$ performance scaling. We suspect that doing even better than the $O(n^2)$ has some relationship with unexplained hardware acceleration.

---

> > > > ### Comment · Reviewer_C23h · 2024-11-27
> > > >
> > > > Many thanks for your reply. I think that the new experiment and the description of convergence by the reviewers is interesting, but given the contribution of this work, I am inclined to keep my score unchanged.

---

### Official Review · Reviewer_jjb1 · 2024-11-04

**Soundness:** 3
**Presentation:** 4
**Contribution:** 3
**Rating:** 6
**Confidence:** 4

**Summary:**

This paper deals with constrained OT problems and the corresponding entropic regularization formulation which could potentially help researchers and practitioners of OT and Machine Learning to arrive at transport plans that have more complex structure than in the unconstrained case. Theoretically, the work is well grounded in existing literature and the authors have made use of multiple recent advancements to design a Sinkhorn-type algorithm to solve constrained OT problems and also have proposed accelerated convergence methods with corresponding bounds. The authors also provide proofs for the various theorems and propositions under certain assumptions. For methods that rely on kernel approximation, prior work do not account for dynamically evolving kernels in the constrained case such as in case of $K=exp(-C\eta)$ but with the authors approach of using a Lyapunov function to characterize the optimization in the variational formulation allows for solving transport problems with evolving kernels.

**Strengths:**

- This paper combines ideas from several previous works and extends them meaningfully in novel ways to solve constrained OT problems under both equality and inequality constraints.
- The novelty lies in the use of Lyapunov function to characterize the optimization procedure to perform the constraint update dual step
- Authors have presented convergence analysis of the proposed Sinkhorn-type optimization procedure with acceleration mechanisms
- Authors have provided decent survey of related literature relating to OT, ML and efficient solvers for constrained and unconstrained OT.
- Authors have provided detailed proofs for theorems and propositions wherever necessary.

**Weaknesses:**

- The numerical experiments are based on (weak) assumptions such as the cost matrix entries being sampled from uniform distribution in case of random assignment problem or the Rademacher distribution in case of Ranking under constraints (appendix A). It is not clear how the proposed algorithm performs when the cost matrix may not conform to simple distributions.
-  Authors could present experiments that would be more relevant to the target ML community.
- Solving large scale problems would help ascertain the claims made by authors about the usability and efficiency of their proposed approach.

**Questions:**

Questions:
- For the Random assignment problem, how would the algorithm behave if cost matrix is not generated through uniform sampling and instead has some other distribution? Do you have any observations or thoughts?
- Sinkhorn supports backpropagation in neural networks owing to matrix/vector operations and hence can be used as part of the loss function or during intermediate steps in computation. Can you comment on whether your approach can be used directly in a similar fashion? Please explain your reaasoning.

Minor suggestions that won't affect the score:
- Line 166/167: “the following” repeated twice in the statement "We summarize the general form of constrained optimal transport by the following the following linear program (LP)"
- Consider citing combinatorial OT solvers that have been proposed that have successfully shown applications in ML to compute partial optimal transports

---

> ### Author Response · Authors · 2024-11-14
> **Reply to the reviewer (Part 1)**
>
> The authors thank the reviewer for the reviewer's time. We would like to address the technical and theoretical concerns raised by the reviewer. We anticipate changes to the manuscript, and so all equation number references and line references will be referring to the pre-discussion version submitted to this conference. If the response provides some clarity and is satisfactory to the reviewer, we kindly suggest that the reviewer consider an increase of the score.
>
> ### Comment 1
>
> *The numerical experiments are based on (weak) assumptions such as the cost matrix entries being sampled from uniform distribution in case of random assignment problem or the Rademacher distribution in case of Ranking under constraints (appendix A). It is not clear how the proposed algorithm performs when the cost matrix may not conform to simple distributions.
> For the Random assignment problem, how would the algorithm behave if cost matrix is not generated through uniform sampling and instead has some other distribution? Do you have any observations or thoughts?*
>
> **Response:**
>
> The authors would like to clarify that the choice of random assignment problem as the considered problem class is due to the fact that random assignments are considered to be harder cases of OT in the unconstrained case, and so the experiment is designed to be a challenge to the algorithm. Intuitively, the random assignment case implicitly models optimal transport problems when the data dimension is large. For example, when the distance function is $c(x, y) = \lvert x - y\vert_2^2$ and the source and target distribution sampled from distributions in very high dimensions, then the entries of the cost matrix one forms would behave very much like i.i.d. random variables.
>
> Random assignment are also typically harder instances numerically. Indeed one can see that the ranking under constraint example converges much quicker. The Pareto front profiling problem considers MNIST images under constrained optimal transport, which is a widely considered example in optimal transport, such as in the Cuturi paper [1]. Algorithm 1 converges quicker than random assignment in the MNIST case under the Pareto front profiling example, though we omit the detailed performance in the MNIST case for simplicity. Overall, Algorithm 1 works very well for simple distributions.
>
> For random assignment problems in the unconstrained case, Aldous [2] shows that the choice of random distribution of each entry doesn't matter so much, but rather the tail behavior of the distribution determines the convergence behavior. For the case of this work, the uniform distribution is chosen as the performance is well-understood and the constraints and thresholds can be set to easily ensure feasibility.
>
>
>
>
> ### Comment 2
>
> *Solving large scale problems would help ascertain the claims made by authors about the usability and efficiency of their proposed approach.*
>
> **Response:**
>
> Algorithm 1 and Algorithm 2 do have a $O(n^2)$ scaling. For comparable problems, when one has $K = 1$ and $L = 1$ (one equality and one inequality constraint) the run-time complexity of Algorithm 1 would take 2-3 times the compute time as that of the Sinkhorn iteration, and so the performance of the Algorithm 1 is quite robust.
>
> To address the reviewer's concern, the authors will show further experiments that of larger random assignment problems with constraints. We will provide an update to the paper before the end of the discussion period.

---

> > ### Author Response · Authors · 2024-11-14
> > **Reply to the reviewer (Part 2)**
> >
> > ### Comment 3
> >
> > *Sinkhorn supports backpropagation in neural networks owing to matrix/vector operations and hence can be used as part of the loss function or during intermediate steps in computation. Can you comment on whether your approach can be used directly in a similar fashion? Please explain your reasoning.*
> >
> > **Response:**
> >
> > From The authors's understanding, the Sinkhorn autodifferentiation occurs through unrolling the computational graph in Sinkhorn's logsumexp operation. We do not know if this can be applied to our case, as backpropagation through linear system inversion is harder and quite unstable.
> >
> >
> >
> > ### Comment 4
> >
> > *Authors could present experiments that would be more relevant to the target ML community.*
> >
> > **Response:**
> >
> > For the main text, the hope of this work is to enable machine learning practitioners to have more flexibility in using OT by incorporating constraints. While the authors can provide some concrete examples of the use of the proposed methodology, the work's main scope will mainly fall into providing a satisfactory answer to the constrained OT problem.
> >
> > One notable remark we would like to add to the general answer is that this work also includes a fast algorithm for a constrained optimal transport setting of partial optimal transport, for which there are extensive ML applications, especially in computer vision [3]. We show that likewise the Siknhorn-type algorithm and the sparse Newton iteration provides great advantage compared to the existing APDAGD algorithm considered in [4].
> >
> >
> > ### References
> >
> > [1] Cuturi, M. (2013). Sinkhorn distances: Lightspeed computation of optimal transport. Advances in neural information processing systems, 26.
> >
> > [2] Aldous, D. J. (2001). The zeta(2) limit in the random assignment problem. Random Structures \& Algorithms, 18(4), 381-418.
> >
> > [3] Chapel, L., Alaya, M. Z., & Gasso, G. (2020). Partial optimal tranport with applications on positive-unlabeled learning. Advances in Neural Information Processing Systems, 33, 2903-2913.
> >
> > [4] Nguyen, A. D., Nguyen, T. D., Nguyen, Q. M., Nguyen, H. H., Nguyen, L. M., & Toh, K. C. (2024, March). On partial optimal transport: Revising the infeasibility of sinkhorn and efficient gradient methods. In Proceedings of the AAAI Conference on Artificial Intelligence (Vol. 38, No. 8, pp. 8090-8098).

---

> > > ### Author Response · Authors · 2024-11-22
> > > **Update on numerical experiment**
> > >
> > > We are delighted to inform the reviewer that we have updated the manuscript on a larger random assignment instance. For the problem size of $n = 5000$, the proposed Sinkhorn-type and Sinkhorn-Newton-Sparse algorithms were able to converge to the entropically optimal solution in machine accuracy. The experiment shows that the proposed algorithm can scale to very large assignment problems.
> > >
> > > ## Result summary
> > > The Sinkhorn-type algorithm takes 40 iterations and ~80 seconds to converge. The Sinkhorn-Newton-Sparse (SNS) algorithm takes 21 iterations and ~45 seconds to converge. The algorithms are able to achieve good accuracy relatively quickly. In contrast, the case of $n=500$ takes 4 seconds for the SNS algorithm to converge. Thus, we show that the numerical performance even outperforms over the $O(n^2)$ performance scaling. We suspect that doing even better than the $O(n^2)$ has some relationship with unexplained hardware acceleration.

---

### Meta-Review · Area_Chair_Ksht · 2024-12-17

**Metareview:**

The paper presents a novel Sinkhorn-type algorithm for constrained optimal transport problems, extending existing OT formulations to include both equality and inequality constraints. While the theoretical foundations are sound, reviewers highlighted weaknesses. A major concern is the lack of motivation and clear exemplification of the method’s relevance to machine learning, particularly its ability to provide new insights or practical improvements for ML applications involving OT. The experimental setup relies on toy scenarios and fails to demonstrate scalability or applicability to real-world ML tasks, such as domain adaptation or large-scale data alignment. Furthermore, the theoretical analysis of computational complexity is incomplete, and the algorithm’s scaling properties in terms of accuracy remain unclear. Despite revisions, these issues persist, with reviewers noting that the contributions—while technically valid—are incremental and primarily extend previous works without providing sufficient broader impact or practical utility for the ICLR community.

**Additional Comments On Reviewer Discussion:**

During the rebuttal period, reviewers raised concerns about the limited experimental validation, weak theoretical insights into scaling laws, and the lack of motivation for the constrained OT framework. The authors responded with additional experiments and clarifications but failed to address the broader applicability and relevance to ML convincingly. These unresolved issues ultimately weighed heavily in the final decision to reject the paper.

---

### Decision · Program_Chairs · 2025-01-22

Reject